# Distinct contributions of tensile and shear stress on E-cadherin levels during morphogenesis

Girish R. Kale [1,2], Xingbo Yang[3,4], Jean-Marc Philippe[1], Madhav Mani[3], Pierre-François Lenne [1] & Thomas Lecuit [1,5]

During epithelial morphogenesis, cell contacts (junctions) are constantly remodeled by mechanical forces that work against adhesive forces. E-cadherin complexes play a pivotal role in this process by providing persistent cell adhesion and by transmitting mechanical tension. In this context, it is unclear how mechanical forces affect E-cadherin adhesion and junction dynamics. During *Drosophila* embryo axis elongation, Myosin-II activity in the apico-medial and junctional cortex generates mechanical forces to drive junction remodeling. Here we report that the ratio between Vinculin and E-cadherin intensities acts as a ratiometric readout for these mechanical forces (load) at E-cadherin complexes. Medial Myosin-II loads E-cadherin complexes on all junctions, exerts tensile forces, and increases levels of E-cadherin. Junctional Myosin-II, on the other hand, biases the distribution of load between junctions of the same cell, exerts shear forces, and decreases the levels of E-cadherin. This work suggests distinct effects of tensile versus shear stresses on E-cadherin adhesion.

[1] Aix Marseille Université, CNRS, IBDM-UMR7288, Turing Center for Living Systems, 13009 Marseille, France. [2] National Center for Biological Sciences, GKVK Campus, Bellary Road, Bangalore 560065, India. [3] Northwestern University, 2145 Sheridan Road, Evanston, IL 60208, USA. [4] Department of Molecular and Cellular Biology, Harvard University, Cambridge, MA 02138, USA. [5] Collège de France, 11 Place Marcelin Berthelot, 75005 Paris, France. Correspondence and requests for materials should be addressed to P.-F.L. (email: pierre-francois.lenne@univ-amu.fr) or to T.L. (email: thomas.lecuit@univ-amu.fr)

Tissue scale morphogenetic movements are driven by the dynamic remodeling of cell–cell adhesion and contractile actomyosin cytoskeleton at cell interfaces[1–4]. E-cadherin-based cell adhesion machinery is not uniformly distributed at the cell interfaces. Within adherens junctions, E-cadherin forms *cis-* and *trans*-homophilic clusters whose size, density, and lateral mobility (flow) depend, in part, on coupling to F-actin[5–7]. E-cadherin cell adhesion complexes are physically linked to the actomyosin cytoskeleton by α-Catenin and Vinculin, two F-actin-binding proteins[8–13]. Such coupling to F-actin is essential for determining E-cadherin cluster size and number, underlying adhesion maturation, cell–cell cohesion[14,15], epithelial integrity in vivo[9] and cell sorting behavior[16]. Importantly, E-cadherin coupling to F-actin via α-Catenin is dependent on force: α-Catenin's interaction with F-actin can be modeled as a two-state catch bond, where force shifts the complex to a strongly bound-state[11] potentially by a tension-induced conformational change[10,17–20]. This argues that actomyosin-generated tension reinforces coupling to E-cadherin complexes in vivo[21,22]. By virtue of trans-homophilic interactions, E-cadherin complexes transmit these tensile forces across actomyosin cortices of neighboring cells[16]. The link between actomyosin contractility and E-cadherin may promote the regulation of cell adhesion by actomyosin contraction during tissue morphogenesis, though this possibility has not yet been directly addressed in a developmental context.

E-cadherin-based cell adhesion plays a dual role by both maintaining tissue cohesion and by facilitating tissue remodeling under biochemical and mechanical regulation[23–26]. Contractile forces can affect cell adhesion, as they can directly influence the recruitment or turnover of E-cadherin molecules[27]. However, the evidence is sometimes contradictory, in some instances tension stabilizes E-cadherin, while in others tension appears to have the opposite effect. Mammalian cell culture experiments have demonstrated that cells respond to cell extrinsic tensile forces through local reorganization of F-actin cytoskeleton and increased recruitment or stabilization of E-cadherin[12,13,18,20]. However, other experiments have demonstrated that E-cadherin levels are reduced due to signaling downstream of Src and that the contractile activity of Myosin-II is the transducer of this reduction[28]. In addition, higher junctional tension correlates with increased turnover rate of E-cadherin molecules in MDCK cells[29], which in turn depends on the endocytosis/exocytosis of E-cadherin[30], arguing for tension reducing E-cadherin levels. Whether mechanical load regulates E-cadherin-based adhesion in vivo has been comparatively little explored[31,32].

We addressed this question during the early development of *Drosophila* embryonic ectoderm, which undergoes convergent-extension movements[33]. These movements rely on cell intercalation, which involves disassembly of junctions oriented along the dorsal–ventral axis (DV, vertical junctions) of the embryo, followed by elongation of new junctions along the anterior–posterior axis (AP, transverse junctions)[34–36]. Ectodermal cells have two distinctly regulated pools of Myosin-II that are responsible for persistent junction shrinkage as well as elongation[36–38]. First, a pulsatile pool of Myosin-II (medial Myosin-II) produces semi-periodic contractions in the apical cortex, and the pool of Myosin-II in the junctional cortex (junctional Myosin-II) produces anisotropic contractions along vertical junctions. Junctions experience tension of distinct orientation due to Myosin-II contractions in these different actomyosin pools (Supplementary Fig. 1A). Indeed, junctional actomyosin ablation causes relaxation along the axis of the junction[37,39,40] indicating that tension is parallel to the junction (Supplementary Fig. 1B). Medial actomyosin ablation causes relaxation of the junction away from the ablation, perpendicular to the axis of the junction[36,41], indicating that the tension is normal to the junction (Supplementary Fig. 1C). It is believed that these two pools cumulatively generate polarized tension at cell junctions, such that vertical junctions are under greater tension than the transverse junctions. Indeed, tension at cell junctions is planar polarized as measured by laser ablation of actomyosin cortices[39] and by optical tweezers[42]. However, the measured tension is defined at the scale of a whole junction. It is unclear how this junction-level tension translates at the level of E-cadherin molecules to which actomyosin networks are coupled. Further, it is unknown whether the contractions of the medial and junctional actomyosin networks are transmitted to E-cadherin molecules in a different way. Lastly, it remains unresolved whether the medial and junctional actomyosin networks impact differently on E-cadherin recruitment.

In this study, we have investigated the effect of actomyosin contractility on cell adhesion, through the analysis of the load exerted onto E-cadherin. Based on previous studies, Myosin-II activation by phosphorylation of its regulatory light chain can be directly inferred from its recruitment[43]. Thus, we use changes in Myosin-II recruitment as a proxy for changes in its activation and for the changes in the generation of tensile forces themselves. Myosin-II phosphorylation depends on the kinase Rok, which is activated by the small GTPase Rho1. Medial activation of Rho1 depends on Gα12/13 (also called *Concertina*) and its molecular effector, the GEF RhoGEF2[38]. Thus Gα12/13 and RhoGEF2 control medial apical actomyosin tension by specifically regulating apical actomyosin recruitment. We analyze the contribution of medial and junctional actomyosin networks to the load on adhesion complexes and to the recruitment of E-cadherin during morphogenesis of the embryonic ectoderm. Our analysis leads us to consider the differential role of tensile and shear stresses exerted respectively by the medial and junctional actomyosin networks.

## Results

**α-Catenin recruits *Drosophila* Vinculin in adhesion complexes**. In mammalian cells, Vinculin is recruited at E-cadherin adhesion complexes via its binding with α-Catenin[10,12,19]. We asked whether a similar phenomenon occurs in *Drosophila* embryonic ectoderm. We first verified that Vinculin is a component of E-cadherin-based adhesion complexes. Figure 1a shows E-cadherin clusters (arrowheads) co-localizing with Vinculin clusters. This point is qualitatively supported by the similarities in the intensity profiles for Vinculin and E-cadherin in a zoom-in view of a junction (Fig. 1b). To test whether α-Catenin is required for the recruitment of Vinculin to adhesion complexes, we injected embryos with double-stranded RNA (dsRNA) to achieve an RNAi mediated knockdown of α-Catenin (see Methods). α-Catenin knockdown significantly reduced Vinculin density at cell–cell contacts (Fig. 1c, d), implying that α-Catenin is the primary interactor of Vinculin and facilitates Vinculin recruitment in adhesion complexes. We observed that Vinculin was enriched in the apico-lateral domain of ectodermal cells similar to E-cadherin and Myosin-II (Supplementary Fig. 1D–G). With these observations, we conclude that Vinculin is a bona fide component of adhesion complexes in *Drosophila*, similar to its mammalian homologs.

Further, Vinculin was enriched on vertical junctions compared to transverse junctions in the embryonic ectoderm (Supplementary Fig. 1N, O). This distribution strikingly mirrored that of Myosin-II (Supplementary Fig. 1L, M), which is known to be planar polarized[34,44,45]. This is remarkable since E-cadherin, however, is present at a lower concentration on vertical junctions[35,41]. Thus, Vinculin distribution was opposite to that

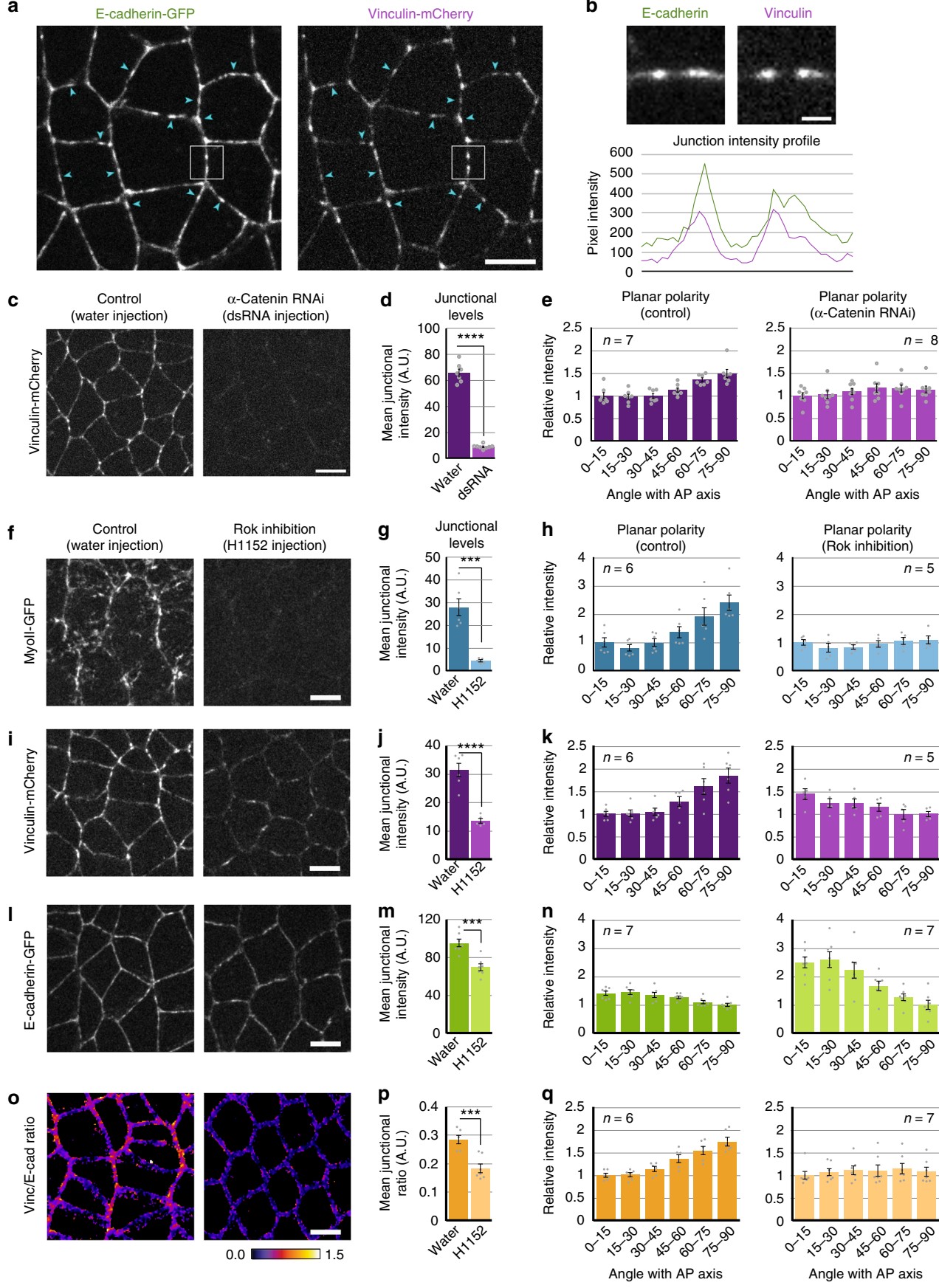

**Fig. 1** Regulation of Vinculin localization. **a** Colocalization between E-cadherin and Vinculin. Various E-cadherin clusters (arrowheads) colocalize with those of Vinculin. **b** Top panels, zoom-in view of boxed junction in **a**. Bottom panel shows the similarities in the intensity profiles for Vinculin and E-cadherin. **c** Representative images showing the distribution of Vinculin in water-injected embryo (left) and α-Catenin dsRNA injected embryo (right). **d, e** Quantifications showing a reduction in Vinculin recruitment and a loss of the planar polarized distribution of Vinculin due to α-Catenin RNAi. **f–q** Rok inhibitor H1152 was injected @ 20 mM concentration to inhibit Myosin-II activity. **f, i, l, o** Representative images showing the distribution of Myosin-II, Vinculin, E-cadherin, and Vinc/E-cad ratio, respectively, in water-injected control embryos (left panels) and H1152-injected embryos (right panels). **g, h** Quantifications showing a reduction in junctional Myosin-II recruitment and a loss of its planar polarized distribution due to Rok inhibition. **i, k** Quantifications showing a reduction in Vinculin recruitment and an inversion of its planar polarized distribution due to Rok inhibition. **m, n** Quantifications showing the reduction in E-cadherin levels and an amplification of its planar polarized distribution due to Rok inhibition. Corresponding representative images and quantifications for changes in Myosin-II distribution are presented in Figure 4a, c, d. **p, q** Quantifications showing a reduction in the Vinc/E-cad ratio and a loss of its planar polarized distribution due to Rok inhibition. Scale bar in **b** is 1 μm. All other scale bars represent 5 μm. All error bars represent SEM. Statistical significance estimated using two-tailed Student's t-test. Images/quantifications in **a–e** and **o–q** come from embryos co-expressing Vinculin-mCherry and E-cadherin-GFP. Junctions marked based on E-cadherin localization. Images/quantifications in **f–k** come from embryos co-expressing Vinculin-mCherry and MyoII-GFP. Junctions marked based on Vinculin localization. Images/quantifications in **l–n** come from embryos co-expressing MyoII-mCherry and E-cadherin-GFP. Junctions marked based on E-cadherin localization. Insets in **e**, **h**, **k**, **n** and **q** indicate number of embryos. ns, $p>0.05$; *, $p < 0.05$; **, $p < 0.01$; ***, $p < 0.001$; ****, $p<0.0001$

of E-cadherin or α-Catenin, which are enriched on transverse junctions relative to vertical junctions (Supplementary Fig. 1H–K). Though, Vinculin planar polarized distribution was lost in the absence of α-Catenin (Fig. 1e). This further indicated that the recruitment of *Drosophila* Vinculin requires α-Catenin. At the same time, these results suggested a differential role for adhesion and contractility in regulating Vinculin recruitment and distribution.

**Myosin-II activity is required for Vinculin enrichment.** The tensile forces generated by Myosin-II are known to produce structural changes in α-Catenin that expose a cryptic binding site for Vinculin and enhance the recruitment of Vinculin to adhesion complexes[10,18,19]. Thus, inhibiting Myosin-II activity can result in a reduction in the junctional recruitment of Vinculin. We tested this idea by injecting in embryos a Rok inhibitor to block the Myosin-II activity (see Methods). Rok inhibition significantly reduced Myosin-II recruitment at junctions and abolished its planar polarity (Fig. 1f–h). The same treatment also reduced Vinculin densities on all junctions (Fig. 1i, j). Noticeably, it inverted the planar polarized distribution of Vinculin, which became similar to that of E-cadherin (Fig. 1k). Rok inhibition also reduced E-cadherin density at junctions and amplified its planar polarity (Fig. 1l–n). Given that Vinculin co-localizes with E-cadherin, we asked if the inversion of Vinculin planar polarity was due to a constitutive localization of Vinculin to E-cadherin in the absence of Myosin-II activity. When we normalized junctional Vinculin density to that of E-cadherin, the Vinc/E-cad ratio, indeed, this ratio was reduced upon Rok inhibition and its planar polarity was lost (Fig. 1o–q) in a manner similar to Myosin-II. The fact that the planar polarity of the Vinc/E-cad ratio qualitatively parallels that of Myosin-II suggests that the recruitment of Vinculin to adhesion complexes is enhanced by Myosin-II activity.

We further tested this by calculating the linear correlation coefficient between junctional Vinculin density and E-cadherin density. The correlation was performed by binning junctions according to their length. We term it the "conditional correlation" (see Methods and Supplementary Fig. 2A). Such a measurement avoids the indirect correlation between the mean junctional densities, as they are proportional to the inverse of junctional length. The correlation was consistently strong in Rok-inhibited embryos independent of junction length (Supplementary Fig. 2B, C), indicating a constitutive association between Vinculin and E-cadherin in the absence of Myosin-II activity. In the presence of Myosin-II activity, the correlation between Vinculin and E-cadherin densities was stronger on shorter junctions

(Supplementary Fig. 2C). This suggests that Myosin-II activity enhances Vinculin recruitment to adhesion complexes at shrinking junctions.

Taken together, these results indicate that Vinculin is recruited to adhesion complexes at low levels independent of Myosin-II activity. In the presence of Myosin-II activity, Vinculin recruitment is enhanced further. In light of these observations, we decided to normalize Vinculin density with that of E-cadherin to specifically focus on the Myosin-II activity-dependent recruitment of Vinculin to E-cadherin.

**The Vinc/E-cad ratio correlates with junctional tension.** We then tested if Vinculin recruitment and Vinc/E-cad ratio are dependent on junctional tension, as this recruitment requires a force-dependent structural changes in α-Catenin. Such function is postulated for *Drosophila* Vinculin[31,46], but has not been demonstrated using explicit tension estimates.

In order to estimate tension distribution, we used and compared two methods of tension estimation; first, "mechanical inference" method[47–49], which uses segmented cell networks to compute relative tensions along junctions within an image (see Methods and Supplementary Fig. 3A); and second, laser ablations method[36,37,39–41,50,51], where post-ablation initial recoil velocity acts as a proxy for tension on junctions (see Methods and Supplementary Fig. 3B). We first applied mechanical inference to our tissue of interest. Figure 2a shows a snapshot from a wild-type embryo where cell junctions are visualized using E-cadherin-GFP signal. Figure 2b shows the corresponding junctional skeleton, on which we implemented mechanical inference. Figure 2c shows the corresponding output of inferred tension from mechanical inference, where the thickness of the junction is proportional to the inferred tension. Note that the mechanical inference captures the tension cables along vertical junctions, which were reported to be under higher tension[39,40]. The planar polarity of inferred tension (Fig. 2d) showed a trend similar to that of junctional Myosin-II and previously described tension distribution.

We further asked which tension estimate, inferred tension or recoil velocity, correlates better with the Vinc/E-cad ratio. We performed laser ablation experiments and found positive and statistically significant correlation between pre-ablation Vinc/E-cad ratio and post-ablation initial recoil velocity (Fig. 2e), although the extent of correlation was low. Vinc/E-cad ratio and inferred tension (Fig. 2f), post-ablation recoil velocity and inferred tension (Supplementary Fig. 3C) show a similar extent of correlation on the same pre-ablation snapshots, indicating that all three tension estimates are comparable. Given that we are pooling

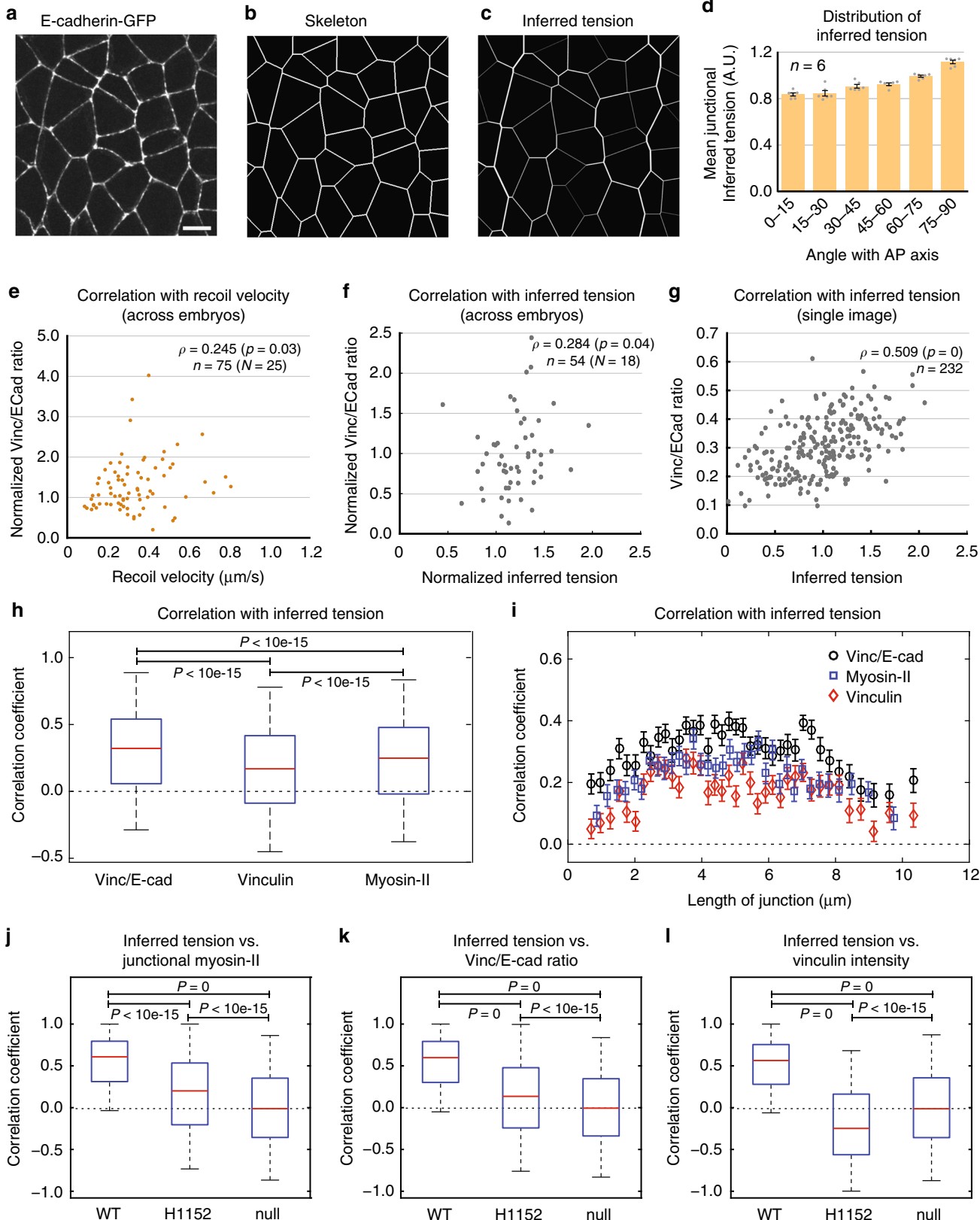

individual data points from different embryos, the low correlation could be an effect of embryo-to-embryo variability. To test this, we plotted the Vinc/E-cad ratio against inferred tension for all junctions from a single snapshot within an embryo and found a strong correlation between Vinc/E-cad ratio and inferred tension

(Fig. 2g). Note that a plot like the one in Figure 2g is difficult to obtain in laser ablation experiments, since laser ablations are performed one junction at a time. Therefore, we decided to use mechanical inference as the primary estimate of tension which yields relative tensions across all images of embryos.

**Fig. 2** Vinc/E-cad ratio correlates with the junctional tension. **a** Representative image used as input for segmentation (scale bar 5 μm). **b** The "skeleton" image with the approximated network of junctions used as input for mechanical inference (see Methods). **c** The output of mechanical inference where line thickness indicates junctional tension. **d** The planar polarized distribution of inferred tension. Number of embryos in inset. **e, f** Scatter plot showing the distribution of the pre-ablation Vinc/E-cad ratio against **e** the post-ablation recoil velocities for multiple events of junctional ablations and **f** the pre-ablation inferred tension, for a subset of junctions in **e** where mechanical inference was feasible. In insets, Spearman correlation coefficient ρ, corresponding p-value, number of ablations events n, pooled from N embryos. **g** Scatter plot showing the distribution of Vinc/E-cad ratio against inferred tension for junctions from a single image. In inset, Spearman correlation coefficient ρ, corresponding p-value, and number of junctions n. **h** Box plots showing the distribution of conditional correlation coefficients for Vinc/E-cad ratio, Vinculin and junctional Myosin-II intensity against inferred tension conditioned on the length of the junction. **i** Same data as **h**, plotted as a function of junction length (also, see Methods). The statistics are based on 37,350 junctions across six embryos. **j–l** Box plots showing the distribution of local correlation coefficients (see Methods) between inferred tension and either the Junctional Myosin-II (**j**, n = 3500 cells), the Vinc/E-cad ratio (**k**, n = 9000 cells) or the Vinculin intensity (**l**, n = 9000 cells), for WT embryos, H1152-injected embryos along with statistical null. Vinculin density and Vinc/E-cad ratio related quantifications come from embryos co-expressing Vinculin-mCherry and E-cadherin-GFP (n = 6 embryos for WT, as well as Rok inhibition), while those for junctional Myosin-II come from embryos co-expression MyoII-mCherry and E-cadherin-GFP (n = 5 embryos for WT and n = 4 embryos for Rok inhibition). The error bars in **d** represent SEM. The boxes in **h** and **j–l** represent 25th to 75th percentiles; the whiskers represent 5th to 95th percentiles; the red lines represent the medians; and p-values estimated using Mann–Whitney U-test. The error bars in **i** represent the standard error across 100 different bins with the same length of junction

Then, we compared the correlations between inferred tension and molecular markers to junction, namely the Vinc/E-cad ratio, Vinculin density and junctional Myosin-II density using mechanical inference. We first performed "conditional correlation" by binning junctions based on their length to avoid artificial correlation induced by variation of junctional length (see Methods and Supplementary Fig. 2D). We found that the Vinc/E-cad ratio correlates better with inferred tension than Vinculin and Myosin-II density (Fig. 2h, i). To avoid spatial variations induced by fluctuations of laser intensities, we also calculated the "local correlation" with inferred tension by binning junctions based on their corresponding cells (see Methods and Supplementary Fig. 2A). In such analysis, the inferred tension strongly correlated with junctional Myosin-II density, as the median local correlation coefficient was 0.6 for wild-type embryos and drops to 0.2 for Rok-inhibited embryos. (Fig. 2j), validating mechanical inference once again. The Vinc/E-cad ratio also showed strong correlation with inferred tension in a manner similar to junctional Myosin-II, as the median correlation coefficient was 0.6 in wild-type embryos and 0.14 after Rok inhibition (Fig. 2k). Further, although the median correlation coefficient between Vinculin density and inferred tension was 0.56 for the wild-type embryos, the correlation was negative (−0.25) for Rok-inhibited embryos (Fig. 2l), which was consistent with the inversion of Vinculin planar polarized distribution upon Rok inhibition (Fig. 1k). With these quantifications we concluded that the Vinc/E-cad ratio strongly correlates with junctional tension in the presence of Myosin-II activity and that the Vinc/E-cad ratio correlates better with tension than Vinculin intensity alone.

To corroborate the results above, we turned to laser ablation experiments again. We found that both junctional Myosin-II density and the Vinc/E-cad ratio showed a statistically significant correlation with recoil velocity (Supplementary Fig. 3D and Fig. 2e). Further, the correlation between recoil velocity and Vinculin density (Supplementary Fig. 3E) was weaker than that between recoil velocity and Vinc/E-cad ratio. Finally, there was no correlation between the recoil velocity and E-cadherin density (Supplementary Fig. 3F), indicating the specificity of the analysis.

Altogether, our data indicated that the distribution of Vinc/E-cad ratio can be used as a ratiometric readout for the distribution of junctional tension.

**The Vinc/E-cad ratio reflects load on adhesion complexes.** Mechanical inference and laser ablations provide an estimate for the junctional tension, a macroscopic quantity that is assumed to be uniform along the junction. E-cadherin adhesion complexes, on the other hand, are distributed in clusters along the junction (Fig. 1a and Supplementary Fig. 1H, J). Adhesion complexes,

composed of E-cadherin, β-Catenin, and α-Catenin, mechanically resist the contractile forces from actomyosin. Thus, adhesion complexes could be under differently oriented contractile forces and resist different magnitude of mechanical loads as they couple independently to the actomyosin network. Vinculin can be an estimate of the mechanical load experienced by each adhesion complex, as individual molecules of Vinculin are recruited to α-Catenin, in a load-dependent manner[17,52]. Given that the Vinc/E-cad ratio correlates with "junctional tension" (a macroscopic quantity), we asked if it can be a readout of the mechanical load at adhesion complexes (a microscopic quantity), potentially providing access to forces at a sub-junctional level.

We address this question by over-expressing E-cadherin to increase its junctional level. The E-cadherin over-expression is expected to reduce the number of Myosin-II molecules per E-cadherin molecule, thereby reducing the load per adhesion complex. The Vinculin level is hence expected to decrease relative to E-cadherin due to a reduction of tension supported by each E-cadherin molecule in the adhesion clusters (Fig. 3a). Indeed, E-cadherin over-expression produced a mild, but significant, increase in its junctional density (Fig. 3b, c), while the distribution of Myosin-II was unchanged (Fig. 3h–j). The junctional tension was also unchanged as shown by recoil velocities after laser ablations (Supplementary Fig. 4A). Concomitantly, there was a reduction in Vinculin density on all junctions (Fig. 3d, e), leading to an even stronger decrease in the Vinc/E-cad ratio (Fig. 3f, g). We suggest that the decrease in Vinculin levels is not due to junctional tension or Myosin-II, as both quantities are unaffected by E-cadherin over-expression, but a response to the decrease of the load per adhesion complex (see Fig. 3a).

Moreover, the Vinc/E-cad ratio can be calculated at a sub-junctional scale, even at a scale as small as individual adhesion clusters. Therefore, we further asked if we can see a consistent change in the Vinc/E-cad ratio at a sub-junctional scale. Pixels in an image are the smallest possible spatial scale available in our analysis. So, we estimated the Vinc/E-cad ratio at individual pixels (see Methods and Supplementary Fig. 5). The Vinc/E-cad ratio was higher at brighter E-cad clusters, suggesting that the mechanical load is inhomogeneous at the sub-junctional level, brighter E-cad clusters bearing larger loads than dim ones. Upon E-cadherin over-expression, we observed a reduction in the Vinc/E-cad ratio across all E-cadherin pixel intensity bins (Fig. 3k). This is consistent with the idea that the load borne by adhesion clusters is dependent on the number of clusters and their sizes (number of molecules per cluster), both of which are known to be dependent on E-cadherin total amount[5].

These quantifications suggest that the Vinc/E-cad ratio at each adhesion complex can be used as a proxy for the "load on

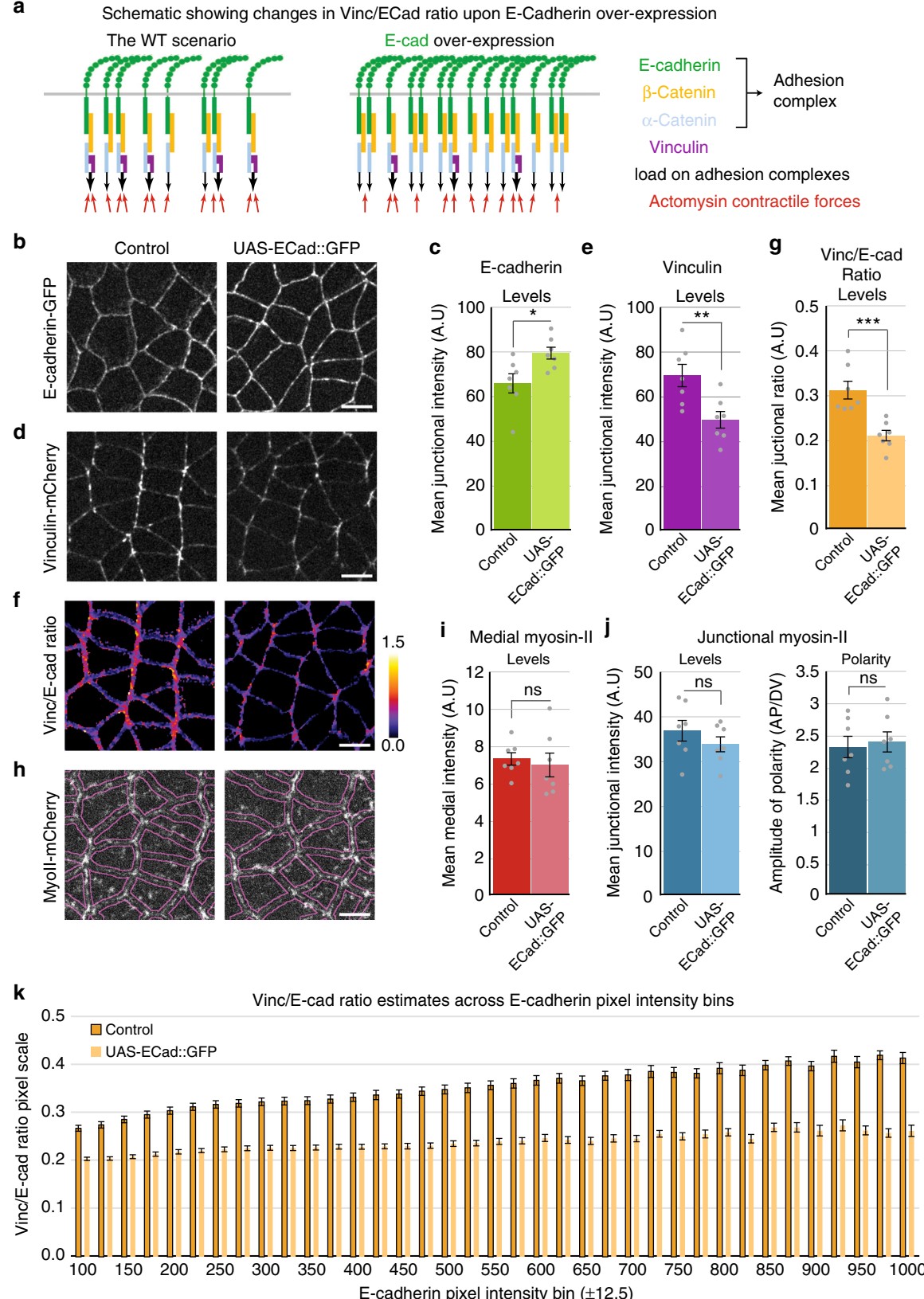

**a** Schematic showing changes in Vinc/ECad ratio upon E-Cadherin over-expression

**k** Vinc/E-cad ratio estimates across E-cadherin pixel intensity bins

adhesion complex". We note that Vinc/E-cad is independent of junctional length, thus can be estimated at a microscopic scale of adhesion clusters and will act as a ratiometric readout of load at E-cadherin adhesion complexes.

**Medial and junctional Myosin-II load adhesion complexes.** E-cadherin adhesion complexes are mechanically coupled to two spatially separated and distinctly regulated pools of Myosin-II, the medial pool and the junctional pool[37,38,43]. The relaxation

**Fig. 3** Vinc/E-cad ratio represents the load on adhesion complexes. **a** Schematics showing the effect of E-cadherin over-expression on Vinculin recruitment. In the WT scenario, Vinculin recruitment is driven by the amount of tension generated by actomyosin contractility loaded on each adhesion complex. After E-cadherin over-expression, the same tension is supported by more adhesion complexes, leading to less Vinculin recruitment, which would result in an overall decrease of Vinc/E-cad ratio. **b, d, f, h** Representative images showing the distribution of E-cadherin (**b**), Vinculin (**d**), Vinc/E-cad ratio (**f**), and Myosin-II (**h**) in genetic outcross control embryos (left panels) and E-cadherin over-expressing embryos (right panels). **c** Quantifications showing an increase in E-cadherin levels at the junctions, quantified as increase in mean junctional intensity. **e** Quantifications showing a decrease in Vinculin levels at the junctions, quantified as a decrease in mean junctional intensity. **g** Quantifications showing a reduction in the mean junctional Vinc/E-cad ratio. **i, j** Quantifications showing that the distribution of Myosin-II has not changed upon E-cadherin over-expression. **k** Quantifications showing Vinc/E-cad ratio estimated at the scale of individual pixels and plotted against corresponding E-cadherin pixel bin intensity. Each bin is 25 intensity units wide. The Vinc/E-cad ratio represents the average of Vinc/E-cad ratio for all pixels in that bin, separately estimated for individual embryos. Mean and SEM are calculated across embryos. All scale bars represent 5 μm. All error bars represent SEM. Statistical significances were estimated using two-tailed Student's t-test. For all quantifications, data come from $n = 7$ embryos for both, control and UAS-E-cad::GFP. Images/quantifications in **b, c** and **h–j** come from embryo co-expressing MyoII-mCherry and E-cadherin-GFP. Images/quantifications in **d–g** and **k** come from embryo co-expressing Vinculin-mCherry and E-cadherin-GFP. In all cases, Junctions/pixels marked based on E-cadherin localization. ns, $p > 0.05$; *, $p < 0.05$; **, $p < 0.01$; ***, $p < 0.001$

kinetics of the actomyosin cortex in laser ablation experiments suggest that medial Myosin-II exerts tension that is predominantly orthogonal to cell contacts, whereas junctional Myosin-II exerts tension that is predominantly parallel to cell contacts[37,39,41]. As the forces produced by these two pools are differently oriented towards the junctions, we asked whether they distinctly load adhesion complexes. First, we inhibited Myosin-II activity globally (Rok inhibition). This treatment reduced the levels of Vinc/E-cad ratio, and suppressed its planar polarity (Fig. 4a–e), consistent with the idea that Myosin-II activity is required to load the adhesion complexes. Next, we tuned the Myosin-II activation in the medial pool only. A recent study demonstrated that a Gα12/13-RhoGEF2-Rho1-Rok pathway phosphorylates and activates Myosin-II in the medial pool downstream of GPCR signaling[38]. Using RhoGEF2-RNAi, we reduced the activation of Myosin-II only in the medial pool, without affecting Myosin-II recruitment in the junctional pool (Fig. 4f, h, i). This treatment decreased the Vinc/E-cad ratio without affecting its planar polarized distribution (Fig. 4g, j). This reduction in the Vinc/E-cad ratio could be due to a reduction of the load generated by the medial Myosin-II or an overall reduction in junctional tension itself. We ruled out the latter possibility by laser ablation experiments (Supplementary Fig. 4B) and the fact that junctional Myosin-II intensity is unchanged in RhoGEF2-RNAi embryos (Fig. 4i). To complement this observation, we increased the recruitment of Myosin-II in the medial pool using Gα12/13 over-expression, without affecting Myosin-II recruitment in the junctional pool (Fig. 4k, m, n). Consistently, this treatment increased the Vinc/E-cad ratio without affecting its planar polarized distribution (Fig. 4l, o).

Thus, a decrease (increase) in the levels of medial Myosin-II decreases (increases) the load on adhesion complexes on all junctions of a cell, as indicated by the decrease (increase) of Vinc/E-cad ratio. In contrast to an overall inhibition of Myosin-II activity, a specific inhibition of medial Myosin-II activity did not affect the planar polarized distribution of the junctional Myosin-II and preserved the planar polarized distribution of Vinc/E-cad ratio (Fig. 4e, j). Thus, we conclude that the planar polarized junctional Myosin-II imposes a larger amount of load on vertical junctions than transverse junctions and determines the planar polarity of the Vinc/E-cad ratio.

**Medial Myosin-II increases junctional E-cadherin density.** Given that medial and junctional pools of Myosin-II load adhesion complexes differently, we asked whether these two pools had distinct impacts on E-cadherin levels at cell junctions. We found that an overall inhibition of Myosin-II activity (Rok inhibition) decreased the E-cadherin density at junctions (Fig. 1l, m). Interestingly, a specific inhibition of the medial Myosin-II using

RhoGEF2-RNAi without perturbation of the junctional Myosin-II (Fig. 4f, h, i) also led to a reduction in E-cadherin levels (Fig. 5a, b) that was comparable to the Rok-inhibited embryos. This indicated that the E-cadherin density is regulated by medial Myosin-II on all junctions, and that the presence of junctional Myosin-II alone did not restore the reduction in the recruitment of E-cadherin. To further test this, we used Gα12/13 over-expression to increase the levels of medial Myosin-II while preserving the levels of junctional Myosin-II (Fig. 4k, m, n). We observed an increase in E-cadherin density at all junctions (Fig. 5c, d). These results suggest that the contractile medial Myosin-II regulates the junctional recruitment of E-cadherin on all junctions.

**Junctional Myosin-II reduces junctional E-cadherin density.** Planar polarized junctional Myosin-II (Supplementary Fig. 1L) is important for junction shrinkage[34,44]. It is hypothesized that the shear stress generated by junctional Myosin-II may stretch the trans-cellular E-cadherin dimers to dissociate them[47], in a manner similar to the detachment of surface-engaged macromolecules due to tangential forces[53], and reduce the stability of cell–cell adhesion. However, this hypothesis has never been tested in vivo with experimental data due to the difficulty of measuring shear stress. Mechanical inference provides a unique way to approximate junctional shear stress from inferred tensions of neighboring junctions, which in turn depends on junctional Myosin-II distribution (Fig. 6a, also see Methods)[47]. Hence, we tested this hypothesis by estimating the "conditional correlation" of E-cadherin density with inferred shear stress on junctions. The shear stress displayed a negative correlation with E-cadherin density on vertical junctions (Fig. 6c, d). In contrast, this correlation reduced when we pooled all junctions with different orientations (Fig. 6b) and vanished for transverse junctions and in Rok-inhibited embryos (Fig. 6e, f). It was interesting to note that the transverse junctions showed the same extent of this correlation, irrespective of Myosin-II activity. This further emphasized that the shear forces were specifically active on vertical junctions. In addition, the correlation with inferred tension was much weaker (Fig. 6b–d). Combined together, these results indicated that shear stress, rather than tension, shows negative correlation with junctional E-cadherin density.

Based on above correlation, we hypothesized that an increase in shear stress would cause a reduction in E-cadherin levels. To test this hypothesis, we used laser ablations to increase junctional shear stress by ablating neighboring junction and checked its effect on the E-cadherin density on the central junction. As shown in the schematic Figure 6g, the shear on the central junction can be increased, if $(T_1 + T_3) > (T_2 + T_4)$ and when we ablate neighboring junction 4 (or alternatively junction 2), as the

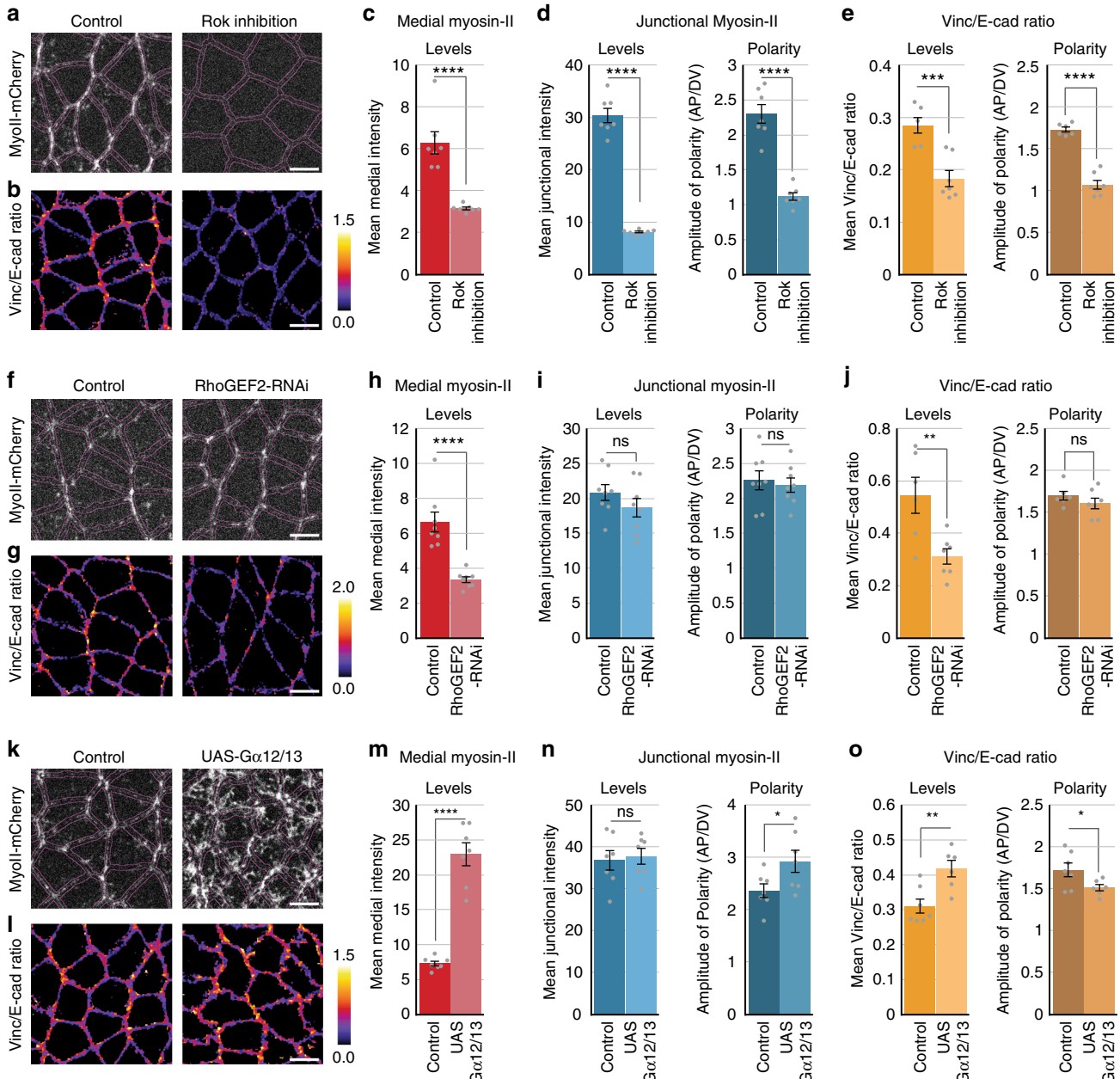

**Fig. 4** Medial and junctional Myosin-II distinctly tunes Vinc/E-cad ratio distribution. Representative images showing the distribution of Myosin-II (**a**) and Vinc/E-cad ratio (**b**), in the water injection "control" embryos (left) and H1152-injected "Rok inhibition" embryos (right). **c**, **d** Quantifications showing the levels of Myosin-II in medial and junction pool, along with planar polarity. Data from n = 7 embryos for both controls and Rok inhibitions. **e** Quantifications showing the mean junctional Vinc/E-cad ratio, along with planar polarity. Data from n = 6 embryos for controls and n = 7 for Rok inhibitions. Representative images showing the distribution of Myosin-II (**f**) and Vinc/E-cad ratio (**g**), in the control embryos (left) and RhoGEF2-RNAi embryos (right). **h**, **i** Quantifications showing the levels of Myosin-II in medial and junctional pool, along with planar polarity. Data from n = 8 embryos for both RhoGEF2-RNAi and control. **j** Quantifications showing the mean junctional Vinc/E-cad ratio along with planar polarity. Data from n = 6 embryos for control and n = 7 embryos for RhoGEF2-RNAi. Representative images showing the distribution of Myosin-II (**k**) and Vinc/E-cad ratio (**l**), in the control embryos (left) and Gα12/13 over-expressing embryos (right). **m**, **n** Quantifications showing the levels of Myosin-II in medial and junctional pool along with planar polarity. Data from n = 7 embryos for both Gα12/13 over-expression and control. **o** Quantifications showing the mean junctional Vinc/E-cad ratio along with planar polarity. Data from n = 7 embryos for control and n = 6 embryos for Gα12/13 over-expression. All scale bars represent 5 μm. All error bars represent SEM. Statistical significance estimated using two-tailed Student's t-test. Images/quantifications in **a**, **c**, **d**, **f**, **h**, **i**, **k**, and **m**, **n** come from embryo co-expressing MyoII-mCherry and E-cadherin-GFP, while those in **b**, **e**, **g**, **j**, **l**, and **o** come from embryo co-expressing Vinculin-mCherry and E-cadherin-GFP. The proportion of tagged vs untagged protein pool varies across different experiments (see Methods: Fly lines and genetics). In all cases, Junctions marked based on E-cadherin localization. Images/quantifications in **b** and **e** are from the same set of embryos as those presented in Figure 1o–q. ns, p > 0.05; *, p < 0.05; **, p < 0.01; ***, p < 0.001; ****, p < 0.0001

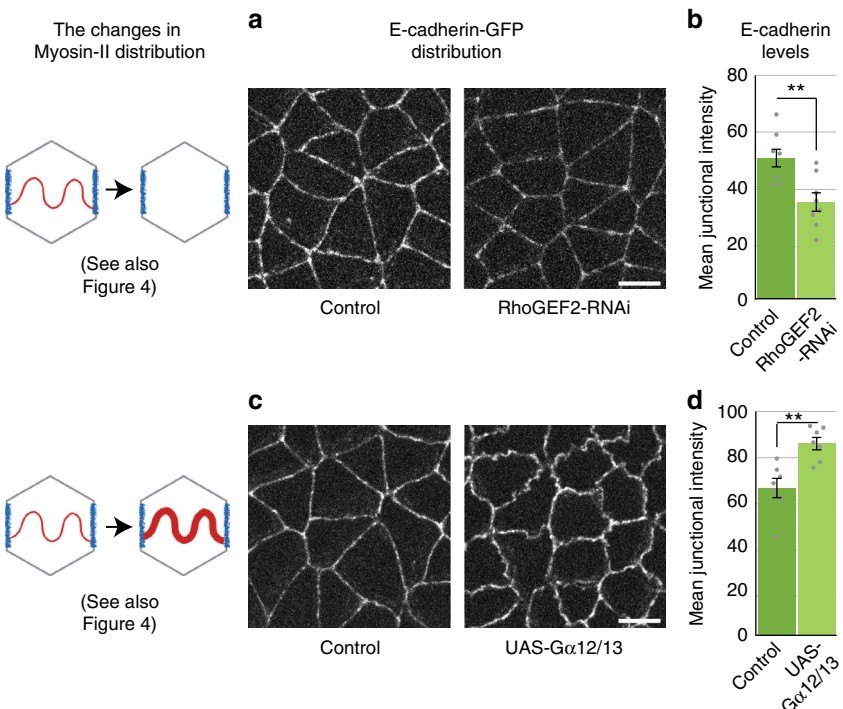

**Fig. 5** Medial Myosin-II tunes the levels of junctional E-cadherin. **a** Representative images showing the distribution of E-cadherin, in the genetic outcross control embryos (left) and RhoGEF2-RNAi embryos (right). **b** Quantifications showing the reduction in E-cadherin levels, quantified as mean junctional intensity. Data from $n = 8$ embryos for both RhoGEF2-RNAi and control. **c** Representative images showing the distribution of E-cadherin, in the genetic outcross control embryos (left) and G$\alpha$12/13 over-expressing embryos (right). **d** Quantifications showing an increase in E-cadherin levels, quantified as mean junctional intensity. Data from $n = 7$ embryos for both G$\alpha$12/13 over-expression and control. All scale bars represent 5 μm. All error bars represent SEM. Statistical significance estimated using two-tailed Student's $t$-test. Images/quantifications in all panels come from embryo co-expressing MyoII-mCherry and E-cadherin-GFP, though the proportion of tagged vs untagged protein pool varies across different experiments (detailed in Methods: Fly lines and genetics). Corresponding Myosin-II related images/quantifications are presented in Figure 4. In all cases, Junctions marked based on E-cadherin localization. ns, $p > 0.05$; *, $p < 0.05$; **, $p < 0.01$; ***, $p < 0.001$

ablation releases the tension on the neighboring junction and enhances the asymmetry of neighboring tensions. We performed mechanical inference on pre-ablation time point for several instances of laser ablations and identified post hoc the ablation events ($n = 47$) where we had ablated either the junction 2 or 4. Then, we estimated the changes in E-cadherin density for the central junction, over 20 s post-ablation. Strikingly, we found that the E-cadherin density reduced for junctions that experienced an increase in shear stress (Fig. 6h). Further, the shear induced reduction in E-cadherin density was in contrast to an average increase in E-cadherin density in all junctions over the same time interval. Also, the reduction in E-cadherin density could not be attributed to a dilution effect, as the junctions of interest actually shrank (Fig. 6h inset). Thus, an increase in shear reduced E-cadherin density. Combined together, these results argue that shear stress enhances the dissociation of E-cadherin on shrinking junctions by shearing the adhesion complexes during junction remodeling.

Together these experiments suggest that the medial Myosin-II increases the levels of junctional E-cadherin by loading the adhesion complexes on all cell junctions, while planar polarized junctional Myosin-II decreases E-cadherin levels by exerting shear forces on the adhesion complexes at vertically shrinking junctions and regulates junction remodeling.

## Discussion
How contractile forces generated by Myosin-II activity regulate junction remodeling during morphogenesis is still an open

question. In this study, we have used Vinculin as a molecular force sensor on E-cadherin complexes, whose recruitment to adhesion complexes is modulated by the contractile activity of Myosin-II and the resulting tensile forces; hence its ratio with E-cadherin provides a potential ratiometric readout of mechanical forces on E-cadherin adhesion complexes at cell junctions. Using mechanical inference and laser ablation, we found that the enrichment of Vinculin relative to E-cadherin can be used to estimate the distribution of load on E-cadherin at cell junctions. With our experiments we compared four quantities, namely Myosin-II intensity, Vinc/E-cad ratio, inferred tension (in mechanical inference analysis) and recoil velocities (in laser ablation experiments). Our analysis shows that these quantities have striking similarities, in terms of what they report. At the same time, each one of them has its own unique features that might carry distinct significance based on what aspect of force generation/transmission/sensing/transduction might be of interest. Only the distribution of Myosin-II can inform about where the tension is generated; only Vinc/E-cad can tell how E-cadherin complexes experience tension; inferred tension, however, is agnostic about the source of tension and reports a cumulative effect of cellular and tissue scale tension; recoil velocities directly report on the physics of the local environment of the ablated junction and can be compared across embryos. Given caveats for each of these methods, the combination of two or more quantities is necessary to get a complete picture of tension distribution as shown here.

While we have established Vinc/E-cad ratio as a ratiometric readout for tension, it remains to be determined how this ratio

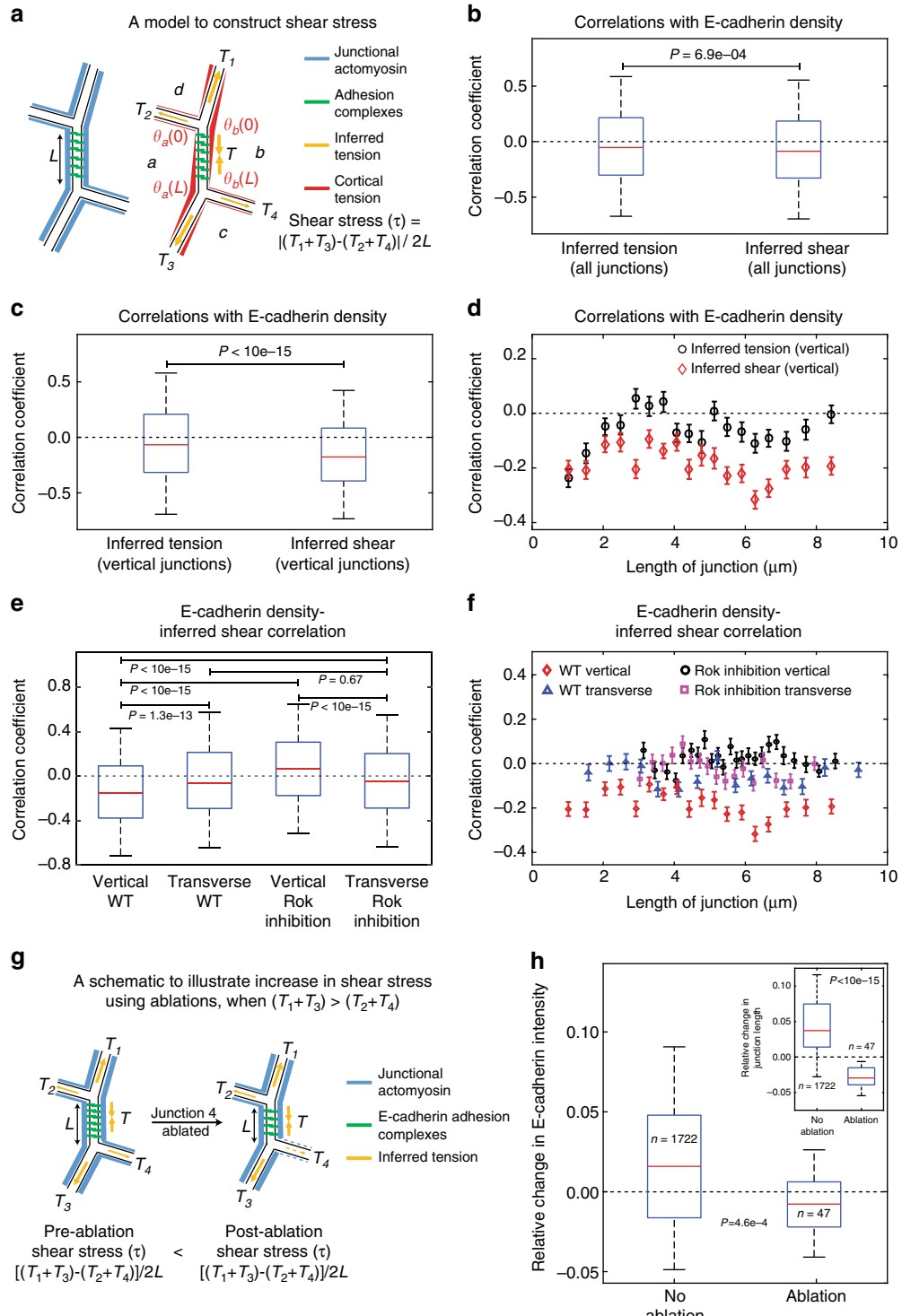

depends on junctional tension explicitly. The conditional correlation revealed a length-dependent correlation between Vinc/E-cad ratio and inferred tension, with reduced correlation for short and long junctions (Fig. 2i). This could suggest a non-linear dependence of Vinc/E-cad ratio on junctional tension with saturated response at short junctions under large tension and the presence of a tension threshold for activation at long junctions under small tension. Experiments with quantitatively controlled tension could determine the response curve of Vinc/E-cad ratio to the magnitude of tension.

Next, we tuned the loading forces on adhesion complexes by increasing the E-cadherin levels and revealed that the Vinc/E-cad ratio can be a load sensor at the adhesion complex scale. Given that the stoichiometry between Vinculin and E-cadherin (proportional to Vinc/E-cad ratio) is a dimensionless quantity, we argue that the Vinc/E-cad ratio estimates the load experienced by individual (diffraction limited) adhesion clusters. It is interesting to note that the distribution of Vinc/E-cad ratio is not homogeneous along a junction (e.g. Figures 1o, 3f, 4b, g, l) and that the Vinc/E-cad ratio is greater at brighter E-cadherin pixels (Fig. 3k).

**Fig. 6** Increase in shear stress reduces the levels of junctional E-cadherin. **a** Schematic showing the model of the junction to construct shear stress. Left schematic shows the distribution of junctional Myosin-II and E-cadherin for central junction. Right panel translates the junctional Myosin-II distribution into inferred tension and asymmetrical cortical tension, which imposes the shear stress on E-cadherin (see Methods: shear stress). **b** Correlation of inferred tension and shear with E-cadherin intensity conditioned on the length of the junction (see Methods). The statistics are based on 19,500 junctions across five embryos. **c**, **d** Correlation of inferred tension and shear with E-cadherin intensity conditioned on the length of the junction, only for the vertical junctions' subset from the data in **b**. **e**, **f** Correlation of inferred shear with E-cadherin intensity conditioned on the length of the junction (see Methods) for vertical and transverse junctions for wild type and Rok-inhibited embryos. The statistics for wild-type (Rok inhibited) embryos are based on 15,000 junctions across 5 embryos (10 embryos), for both transverse and vertical junctions. **g** A model for an increase in junctional shear stress in response to laser ablations. Ablated junction is shown with dashed lines to indicate disrupted actomyosin cortex and reduced junctional tension. **h** Relative changes in E-cadherin intensity and in junction length (in inset), for control junctions (No ablation) and junctions where shear increased due to an ablation in the neighboring junction (Ablation). Relative change = (final value−initial value)/(initial value). We performed bootstrap sampling on 47 junctions with increased shear across 24 ablation events (pooled from 11 embryos) and 1722 control junctions across 24 snapshots, respectively, and averaged over each sample to obtain averaged distribution as plotted. Quantification in all panels come from embryos co-expressing Vinculin-mCherry and E-cadherin-GFP. The error bars in **d** and **f** represent the standard error across 100 different bins with the same length of junction. The boxes in **b**, **c**, **e**, and **h** represent 25th percentiles to 75th percentiles; the whiskers represent 5th percentiles to 95th percentiles; and the red lines represent the medians. *P*-values estimated using Mann–Whitney *U*-test

This indicates that load distribution is inhomogeneous along the junction and that the junctional subdomains with higher E-cadherin density experience greater load. This observation is consistent with a recent study[54], which reported that the mechanosensitive conformational changes in α-Catenin can be observed predominantly in larger E-cadherin clusters.

We used Vinc/E-cad ratio as a load estimate to study the effect of contractile forces from two distinct pools of Myosin-II, the medial and the junctional pool. The two pools are distinct in terms of their upstream regulation and have been studied in the *Drosophila* embryonic ectoderm[38]. They are mechanically coupled to adhesion complexes to exert forces on cell–cell contacts. In this study, we showed that these two pools of Myosin-II have distinct impact on the distribution of load on E-cadherin. Medial Myosin-II is known to produce isotropic contractions and we found that it loads adhesion complexes across all junctions within a cell. In contrast, the planar polarized junctional Myosin-II biases the load towards vertical junctions, thus regulating the planar polarity of load. We have quantitatively demonstrated that both pools of Myosin-II exert forces on E-cadherin complexes and cell contacts.

The load generated due to activity of medial Myosin-II increases the levels of junctional E-cadherin. This observation is consistent with a study in the *Drosophila* mesoderm[32], where it is observed that the activity of medial Myosin-II protects E-cadherin from a Snail-mediated downregulation. A change in junctional Rho signaling can also change E-cadherin levels through its impact on the F-actin organization and Myosin-II activity[55–57]. We think that this is not the case as the junctional Myosin-II levels and presumably junctional Rho signaling is unchanged when we specifically tuned medial Myosin-II. In fact, the changes in junctional E-cadherin levels correlated with the changes in medial Rho signaling downstream of activation by the Gα12/13-RhoGEF2 signaling module. Given that the inhibition of medial Rho signaling (RhoGEF2-RNAi) and Rok inhibition have similar impact on E-cadherin levels, we argue that the effect of medial Rho signaling on E-cadherin levels is through its effect on medial Myosin-II activation. It remains to be determined if the effect of medial Myosin-II activity on the levels of junctional E-cadherin is a mechanosensitive response or not.

We used mechanical inference to study the effect of junctional Myosin-II on junctional E-cadherin levels. We constructed a model to estimate shear stress based on inferred tensions. In this model, an asymmetric distribution of inferred tension on opposite sides of the junction generates shear stress that stretches E-cadherin trans-dimers on shrinking junctions[47] and can destabilize cell adhesion by dissociating E-cadherin trans-homophilic interactions, similar to the surface detachment of macromolecularly-coated beads due to tangential forces[53]. Strikingly, we observed a negative correlation between the inferred shear stress and the junctional E-cadherin levels. Particularly, the negative correlation was specific to vertical junctions (the category to which shrinking junctions belong) and vanished on either the transverse junctions or the junctions from Rok-inhibited embryos. Further, an increase in shear stress in laser ablation experiments demonstrated a causal relationship with reductions in E-cadherin density (Fig. 6g, h). These observations argue that junctional Myosin-II enhances dissociation of E-cadherin on shrinking junctions during junction remodeling via a shear effect on E-cadherin complexes. However, we cannot rule out the possibility that medial Myosin-II also contributes to the inferred shear stress to some extent, as the mechanical inference does not specify the source of the forces.

We hereby propose a mechanical model for cell junction remodeling, where we highlight the importance of the subcellular origin of contractile forces and their mechanical effect, namely tensile vs shear stress, in promoting a change in the levels of E-cadherin at cell contacts and on junction dynamics. The mechanisms that generate the different responses in E-cadherin levels remain unknown. We have established Vinculin as a molecular force sensor, but it remains to be determined whether Vinculin is involved in the stabilization of adherens junctions by regulating E-cadherin levels as a mechanotransducer. Vinculin is not essential for survival in *Drosophila*[58], raising questions about the necessity of its function as a mechanotransducer. To reveal the mechanism by which actomyosin contractility regulates E-cadherin levels, it is essential to study the magnitude and orientation of contractile forces, the spatial distribution of mechanical coupling between the adhesion complexes and the actomyosin network, and the different modes of energy dissipation at adhesive complexes under mechanical forces. Given this distinction between tensile and shear stress in the regulation of E-cadherin at cell contacts, it will also be important to consider the dynamics of E-cadherin complexes at cell contacts as well as at vertices. Vinculin and E-cadherin are present at high levels at vertices. A recent study demonstrated that E-cadherin accumulation at vertices shows oscillatory patterns, which are coordinated with junction shrinkage[59]. Further, this study also shows that vertices exhibit "sliding behavior" during junction shrinkage that is consistent with our report that shear stress remodels adhesive complexes across cell membranes at junctions and, potentially, vertices as well.

We speculate that adhesion mediated by E-cadherin has evolved to stabilize complexes under tensile stress and to

constantly remodel them under shear stress. Tensile (i.e. normal) stresses reinforce cell–cell coupling to induce tissue deformation such as tissue invagination. The shear mode also maintains adhesion but dynamically, thereby allowing tissue remodeling such as during cell intercalation in the ectoderm: on average the density of complexes could remain constant but the turnover of homophilic bonds would be increased. The differential effect of tensile and shear stress on E-cadherin dynamics has the potential of reconciling conflicting evidence on the role of contractile forces on adhesion and to open a study of energy dissipation at E-cadherin adhesion complexes in the study of cell–cell adhesion[4].

## Methods

**Fly lines and genetics**. *Vinculin-GFP* and *Vinculin-mCherry* are fluorescently tagged transgenes of Vinculin. Vinculin gene was tagged at its N-terminus with either superfolder GFP or mCherry, using a pFlyFos025866 Fosmid which encompasses the 8 kb of Vinculin gene along with 23.4 kb upstream and 6.8 kb downstream regions modified by Recombineering[60]. Tagged Fosmids were inserted in the genome at attp2 or attp40 landing sites, respectively using PhiC31-mediated site-specific insertion transgenesis (Transgenesis performed by BestGene, Inc.). *Vinculin-GFP* is used alone to describe Vinculin distribution in the ectodermal cells (Supplementary Fig. 1G, N and O). *Vinculin-mCherry* is always used in combination with either *MyoII-GFP* to quantify Vinculin recruitment (Fig. 1i–k) or *E-cadherin-GFP* to quantify Vinculin recruitment (Figs. 1a–e, 2h, i, l, 3d, e, Supplementary Fig. 2B, C, 3E) to estimate Vinc/E-cad ratio (Figs. 1o–q, 2d–i, k, 3f, g, k, 4b, e, g, j, l, o), and/or to quantify E-cadherin recruitment.

*E-cadherin-GFP* is a homozygous viable DE-cadherin knock-in at the locus[61]. It is either used alone to exemplify E-cadherin distribution in the ectodermal cells (Supplementary Fig. 1E, H and I) or in combination with either *MyoII-mCherry* or *Vinculin-mCherry*. The combination with *MyoII-mCherry* is used to quantify E-cadherin recruitment (Figs. 1l–n, 3b, c and 5a–d), along with Myosin-II. The combination with *Vinculin-mCherry* is used to quantify Vinculin recruitment, to estimate Vinc/E-cad ratio, and/or to quantify E-cadherin recruitment (Figs. 3k, 6b–f, h; Supplementary Fig. 2B, C, 3F, 5B, C).

*MyoII-mCherry* and *MyoII-GFP* are tagged constructs of *Drosophila* "Myosin-II regulatory light chain" encoded by gene *spaghetti squash* (sqh for short) downstream of its native ubiquitously active promoter. Some articles also refer to them as *sqh-mCherry* or *sqh-GFP*. *MyoII-mCherry* is always used in combination with *E-cadherin-GFP* and is used to quantify Myosin-II recruitment (Figs. 2h–j, 3h–j, 4a, c, d, f, h, i, k, m, n). *MyoII-GFP* is either used alone to exemplify its distribution in ectodermal cells (Supplementary Fig. 1F, L and M), to quantify its recruitment (Supplementary Fig. 3D), or in combination with *Vinculin-mCherry* to quantify Myosin-II recruitment (Fig. 1f–h). Gifts from Adam Martin (both on chromosome 2).

*α-Catenin-YFP* is a Cambridge Protein Trap Insertion line (CPTI-002516). DGRC #115551. This is used to describe α-Catenin distribution in ectodermal cells (Supplementary Fig. 1J and K).

*67-Gal4* (mat αTub-GAL4-VP16) is a ubiquitous, maternally supplied, Gal4 driver. This is used in combination either with *MyoII-mCherry* and *E-cadherin-GFP* OR with *Vinculin-mCherry* and *E-cadherin-GFP* OR with *MyoII-GFP* in knockdown/over-expression experiments (see below).

*UAS-ECad::GFP* produces GFP-tagged version of wild-type E-cadherin under UAS promoter. For E-cadherin over-expression, virgin females with the genotype "+; 67-Gal4, *MyoII-mCherry*, *E-cadherin-GFP*;+" (Fig. 3b, c, h–j; Supplementary Fig. 4A) or "+; 67-Gal4, *Vinculin-mCherry*, *E-cadherin-GFP*;+" (Fig. 3d–g, k) were crossed to males with genotype "+; *UAS-ECad::GFP*;+" (or to control males with genotype "y, w;+;+"). Previously used in ref. [41].

RhoGEF2-RNAi was achieved using RhoGEF2 TRiP line (Bloomington #34643). It produces a short-hairpin RNA downstream of a UAS promoter (*UAS-RhoGEF2-shRNA*) that targets RhoGEF2 mRNA to perform RNAi-mediated knockdown. To achieve an effective RNAi during early embryonic development, virgin females with the genotype "+; 67-Gal4, *MyoII-mCherry*, *E-cadherin-GFP*;+" (Figs. 4f, h, i and 5a, b) or "+; 67-Gal4, *Vinculin-mCherry*, *E-cadherin-GFP*;+" (Fig. 4g, j) or "+; 67-Gal4, *MyoII-GFP*;+" (Supplementary Fig. 4B) were first crossed to males with genotype "+;+; *UAS-RhoGEF2-shRNA*" (or control males with genotype "y, w;+;+"). F1 virgins from these crosses were further out-crossed to males with genotype "y, w;+;+".

*UAS-Gα12/13* produces untagged version of wild-type Gα12/13, which is the α-subunit of the heterotrimeric G-protein complex that associates with GPCR *smog*[38]. For Gα12/13 over-expression, virgin females with the genotype "+; 67-Gal4, *MyoII-mCherry*, *E-cadherin-GFP*;+" (Figs. 4k, m, n and 5c, d) or "+; 67-Gal4, *Vinculin-mCherry*, *E-cadherin-GFP*;+" (Fig. 4l, o) were crossed to males with genotype "+; *UAS-Gα12/13*;+" (or to control males with genotype "y, w; +; +"). Gift from Naoyuki Fuse.

**Embryo preparation, RNAi, and drug injections**. Embryos were prepared as described before[34,62,63]. Briefly, embryos were dechorionated using bleach, for

about 40 s and then washed thoroughly with distilled water. The embryos were then aligned on a flat piece of agar and then glued to a glass coverslip. These embryos can be submerged in water and can be imaged directly. Alternatively, glued embryos were kept in an airtight box containing Drierite for about 7 min, then covered in halocarbon oil, and then injected with RNase-free water containing either dsRNA or drugs.

α-Catenin RNAi (Fig. 1c–e) was achieved by injecting dsRNA in embryos, as previously described[9]. Briefly, dsRNA probes against α-Catenin were made using PCR products containing the sequence of the T7 promoter targeting nucleotides 101–828 of α-Catenin sequence (GenBank accession D13964). dsRNA prepared (as already described) were diluted for injection at 5 μM concentration and injected within the first hour of embryonic development to achieve maximum knockdown. As a control, separate set of embryos of the same stage were injected with similar volume of RNase free water.

Rok inhibition (Figs. 1f–q, 2j–l, 4a–e and 6e, f; Supplementary Fig. 2B, C) was achieved through drug injections. H1152 is a membrane permeable pharmacological inhibitor that has high specificity for Rok and blocks its kinase activity. This drug was dissolved in RNase-free water @ 20 mM and injected at the end of cellularization. As a control, separate set of embryos of the same stage were injected with similar volume of RNase-free water. The reduction in Myosin-II recruitment acts as a direct readout of Rok inhibition. The effect of Rok inhibition on Vinc/E-cad ratio could not be assessed directly, as Myosin-II could not be imaged simultaneously. Thus, first the inhibition was performed in embryos expressing *MyoII-GFP* and *Vinculin-mCherry*. The reduction in Vinculin intensity can then be used to assess the extent of Rok inhibition in embryos expressing *Vinculin-mCherry* and *E-cadherin-GFP*, while also estimating the Vinc/E-cad ratio. The reduction in E-cadherin was cross-checked with another set of embryos expressing *E-cadherin-GFP* and *MyoII-mCherry*, where reduction in *MyoII-mCherry* recruitment acted as a direct readout of Rok inhibition. This is an internally reproduced experiment, as the same treatment (H1152 injection) was performed on three different sets of embryos with distinct genotypes, along with respective (water injection) controls.

**Imaging**. Time-lapse images were acquired to encompass stage 7 to 8 of the embryonic development[64], which needs ~15 min at room temperature (~22 °C). Embryos were imaged for 20–30 min depending on the experiment, on a Nikon spinning-disk Eclipse Ti inverted microscope using a ×100 1.45 NA oil immersion objective. MyoII-mCherry and E-cadherin-GFP signals were captured every 30 s or higher, on 11 Z-planes, separated by 0.5 μm. Vinculin-mCherry and E-cadherin-GFP signals were captured every 30 s or higher, on 7 Z-planes, separated by 0.5 μm. A Nikon spinning-disc Eclipse Ti inverted microscope using a ×100 1.4 NA oil immersion objective was used for imaging α-Catenin-YFP. Both systems acquire images using the MetaMorph software. Laser power and exposure settings had to be optimized separately for each experiment, as the fraction of fluorescently tagged vs untagged protein pool changes, in accordance with changes in the maternal and zygotic genotype (see Methods: Fly lines and Genetics). In all cases, imaging conditions were optimized to get best signal while minimizing photo-bleaching, and were kept identical between control and perturbation embryos.

**Laser ablation experiments**. Ablations were performed in a 10-min time window around stage7b (stage7b ± 5 min) on an inverted microscope (Eclipse TE 2000-E; Nikon) equipped with a spinning-disc (Ultraview ERS, Perkin Elmer) for fast imaging. Time lapse at a single z-plane was acquired using a ×100 1.4 NA oil immersion objective. Two color images were acquired in sequence on the same camera, when necessary. Ablations were performed in parallel with image acquisition. Ablation events were obtained by exposing the junctions, for duration of 2–3 ms, to a near-infrared laser (1030 nm) focused in a diffraction-limited spot. Laser power at the back aperture of the objective was ~800 mW.

**Image analysis and statistics**. All image processing was done using FIJI freeware. Raw images were processed using a custom written macro. First, it generated a "signal image" by using the StackFocuser plugin to determine the plane of best focus, followed by a maximum-intensity projection of only three z-planes (one z-plane in focus determined by StackFocuser + two z-planes basal to it). The macro also generated a "background image", first, using a maximum-intensity projection of basal-most three planes, followed by applying a 50 pixel radius median filter. The macro then subtracted the "background image" from "signal image" to produce "processed image". Supplementary Figure 6A–D exemplify the output of this workflow.

The images were independently segmented using "Packing analyzer v2.0" (described in ref. [65]), which was implemented as a plugin in FIJI, to get segmented junctional networks. E-Cadherin, Vinculin, Myosin-II, or α-Catenin intensities were used, depending on the genotype of the embryos (and in that order of preference), for image segmentation in order to identify cell–cell contacts in a semi-automated manner (exemplified in Supplementary Fig. 6E). Using another custom written FIJI macro, the segmentation was used to demarcate the junctional ROIs of about 5-pixel width, such that the vertices (tri-cellular junctions) are excluded (exemplified in Supplementary Fig. 6F). Line densities were measured to calculate "mean junctional intensity". Junctions were categorized based on their angle

relative to AP axis into six "angle bins" (0–15, 15–30… 75–90°). An average of the junctional line densities was calculated within each "angle bin" to get six values of "Averaged Line Density (ALD)" for every embryo. A further average of these ALD values acts as a data point per embryo to estimate "mean junctional intensity". The ratio between the ALD for "0–15" (AP) and "75–90" (DV) categories produces the "Planar Cell polarity (PCP)" value per embryo as either AP/DV or DV/AP as mentioned in the Y-axis labels for respective bar plots in the manuscript. A further average of these PCP values produces "amplitude of polarity". Alternatively, the planar polarity of a protein was represented as "relative intensity", where the ALD values from the "angle bins" were normalized by the ALD of either "0–15" or "75–90" category, whichever is smaller. In case of Vinc/E-cad ratio, similar calculations were performed after having calculated the ratio between Vinculin and E-cadherin line densities for every junction. The same segmentation was also used to identify the medial ROIs (exemplified in Supplementary Fig. 6G) which were at least 2 pixels away from any junctional ROI and tri-cellular junctions, and were specifically used for medial Myosin-II intensity estimates. An average of medial area densities (averaged area density) was calculated to get one data point per embryo to estimate "mean medial intensity" for Myosin-II across multiple embryos.

In case of ablation experiments, images were first processed using the "rolling ball" background subtraction method implemented in FIJI (rolling ball radius 50). Junctional ROIs were drawn manually (5 pixels wide) on the ablated junction and on 20+ neighboring junctions. Then, the "line density" for the ablated junction was divided by the average of the line densities for the neighboring junctions. This yielded the "normalized junctional intensity". Such normalization is necessary to reduce embryo-to-embryo variability. In case of Vinc/E-cad ratio, similar calculations were performed after having calculated the ratio between Vinculin and E-cadherin line densities for all marked junctions. The vertices of the ablated junction were tracked manually to estimate the recoil velocity in 2 s after the ablation. Spearman correlation gave an estimate of the extent of correlation between "pre-ablation normalized junctional intensity" and corresponding "post-ablation initial recoil velocity".

For "pixel scale analysis" of Vinc/E-cad ratio, we identified E-cadherin-positive pixels by estimating the signal-to-noise ratio (SNR) at all pixels (Supplementary Fig. 5A, B) and measured Vinc/E-cad for the pixels with SNR > 1. We empirically decided the range of E-cadherin pixel intensities to span an order of magnitude, such that each intensity range hosts statistically meaningful number of pixels (Supplementary Fig. 5C).

The "mean values" and "standard errors on mean" were calculated from "n" data points. The same data points were used for testing statistical significance. In planar polarity and junctional intensity measurements, "n" is the number of embryos. Error bars indicate SEM (i.e., SD/√n). The p-values were estimated using Student's t-test, wherever applicable. In laser ablation experiments, "n" is the number of ablated junctions that are pooled from many embryos and the p-values were estimated using Mann–Whitney U-test (Supplementary Fig. 4A, B).

In case of correlational analysis, correlation coefficients are calculated from "n" data points, pooled from many embryos. We have used either the Spearman or Pearson correlation. Spearman correlation looks for monotonic relationships, thus allowing relationships to be non-linear as well. Pearson correlation, in contrast, looks for linear relationships. In our analyses, the differences between Spearman correlation coefficients and Pearson correlation coefficients are minimal, if any, indicating that most relationships are linear. Though, we have used Spearman correlation when the number of data points is small (<100), where an assumption of linear relationship might not be justified due to fewer data points.

All measurements were performed on 4–25 embryos spread over at least three rounds of embryo collection and preparation. The sample sizes were not predetermined using any statistical methods. The experiments were not randomized, and the investigators were not blinded to allocation during experiments and outcome assessment.

**Mechanical inference**. Mechanical inference is an image-based force inference technique that takes a segmented cellular network as the input and estimates relative tensions along cell junctions by assuming force balance at each vertex[47–49,66,67]. Force balance at vertices can be a good assumption even for dynamic tissues like *Drosophila* embryonic ectoderm, when junctional tensions are much larger than unbalanced residual forces at cell vertices. This is further evidenced by the observation that recoil velocity of laser ablated junction is much larger than the migration velocity of cell vertex during cell intercalation associated with germband extension. Arguably, the morphogenetic movement is driven by the small unbalanced residual forces, but at a much longer time scale compared to the kinetics of cytoskeletal components. Indeed, the time scales of junction shrinkage/extension (in the order of 100 s) are at least an order of magnitude longer than the time scales corresponding to the on/off kinetics of cytoskeletal components through activation/inactivation, as well as the turnover of adhesion complexes (in order of 1–10 s). Such separation of time scales would further justify the assumption that the changes in junction length occur in a quasistatic manner.

We implement mechanical inference on segmented images of cell network based on the E-cadherin channel. We collect 30 images at a time interval of 30 s for each embryo. The E-cadherin channel images are processed using the freeware ilastik for pixel classification. The resulting probability maps of pixels are processed using a customized MATLAB script for cell segmentation using a watershed

algorithm. The mechanical inference is performed on the segmented image by imposing force balance at each vertex of the cell where junctional tensions add up to zero. We assume a homogeneous pressure distribution across the tissue based on the observation that the junctional curvatures are negligible in the ectoderm during the time window of observation, hence pressure does not enter the force balance equation.

The relative junctional tensions are obtained by fitting a tension triangulation network perpendicular to the corresponding cell network (Supplementary Fig. 3A). This is termed the variational mechanical inference as the optimal tension network is obtained by minimizing the energy functional $\Omega$ determined by the inner product of the tension network and the cell network:

$$\Omega = \frac{1}{2} \sum_{<a,b>} \left[ (\vec{\mathbf{Q}}_a - \vec{\mathbf{Q}}_b) \cdot \vec{\mathbf{r}}_{ij} \right]^2 - \frac{\Lambda}{2} \sum_{<a,b>} \left| \vec{\mathbf{Q}}_a - \vec{\mathbf{Q}}_b \right|^2,$$

where $\vec{\mathbf{Q}}_a$ and $\vec{\mathbf{Q}}_b$ are nodes of the tension triangulation as shown in Supplementary Figure 3A and $\vec{\mathbf{r}}_{ij}$ is the cell edge vector connecting vertex $i$ and $j$. $\Lambda$ is the Lagrangian multiplier that constrains the mean magnitude of tension to be one[48]. The tension triangulation network is obtained by choosing the set of $\vec{\mathbf{Q}}$ that minimizes $\Omega$. The magnitude of junctional tension along cell edge $ij$, for example, is then calculated as $T_{ij} = \left| \vec{\mathbf{Q}}_a - \vec{\mathbf{Q}}_b \right|$.

To guarantee the tension network to be a triangulation network, we kept only cells with three-fold vertices, which make up most of the cells in the population. Since mechanical inference yields relative tensions within an image, we normalized the average inferred tension to be one.

**Shear stress**. Shear stress on the E-cadherin clusters was obtained from a microscopic model of the junction (Fig. 6a) as previously described[47]. As illustrated in Fig. 6a, tension of the central junction is decomposed into cortical tensions at $a$-cell side and $b$-cell side of the junction: $T = \theta_a(x) + \theta_b(x)$, where $x$ is the coordinate along the junction. While $T$ is constant along the junction, cortical tensions $\theta_a(x)$ and $\theta_b(x)$ can vary along the junction in opposing gradients (red lines in Fig. 6a) as a result of the transfer of tension from one side of the junction to the other. This transfer of tension generates shear stress on E-Cadherin dimers. Therefore, shear stress at any given point along the junction is defined as the gradient of cortical tension $\tau(x) = \partial_x \theta_a(x) = -\partial_x \theta_b(x)$. The average shear stress along the junction is $\tau = \frac{1}{L} \int_0^L \tau(x)\mathrm{d}x = \frac{\theta_a(L) - \theta_a(0)}{L} = \frac{\theta_b(0) - \theta_b(L)}{L}$. To relate shear stress to junctional tensions as obtained from mechanical inference, we assume that cortical tensions are single-valued, i.e. continuous, at vertices, from which we get the relation:

$$T = \theta_a(0) + \theta_b(0) \quad T_1 = \theta_b(0) + \theta_d(0) \quad T_2 = \theta_a(0) + \theta_d(0),$$

$$T = \theta_a(L) + \theta_b(L) \quad T_3 = \theta_a(L) + \theta_c(L) \quad T_4 = \theta_b(L) + \theta_c(L).$$

We solve equations above to get cortical tensions in terms of junctional tensions and substitute to the equation for shear stress to obtain the final expression of shear stress:

$$\tau = \frac{1}{2L} |(T_1 + T_3) - (T_2 + T_4)|,$$

where $L$ is the length of the junction.

**Local correlation and conditional correlation**. The correlations were performed by binning the junctions according to cell, termed local correlation (Supplementary Fig. 2A), or according to junctional length, termed conditional correlation (Supplementary Fig. 2A). These two types of correlation are special cases of partial correlation, defined as the correlation between two random variables $X$ and $Y$ while holding the third variable $Z$ constant, whose correlation coefficient is given by $\rho_{XY|Z} = \frac{1}{\sigma_{X|Z}\sigma_{Y|Z}} E\left[ \left( X|Z - \mu_{X|Z} \right) \left( Y|Z - \mu_{Y|Z} \right) \right]$ where $\sigma_{X|Z}$ $(\sigma_{Y|Z})$ is the standard deviation of $X$ $(Y)$ at fixed $Z$ and $\mu_{X|Z}$ $(\mu_{Y|Z})$ is the average value of $X$ $(Y)$ at fixed $Z$. The partial correlation removes the spurious correlation between $X$ and $Y$ due to the confounding variable $Z$ which is related to both $X$ and $Y$.

The local correlation avoided the temporal and spatial variations of tension and fluorescence intensity, and yielded the correlation coefficient for each cell. We implemented the local correlation by computing the Pearson correlation coefficient between inferred tension and either the Vinc/E-cad ratio, Myosin intensity, or Vinculin intensity among junctions within each cell. The resulting correlation coefficients were combined across all cells and multiple embryos to yield a distribution as shown in Figure 2j–l.

The conditional correlation avoided the spurious intensity–intensity and intensity–tension correlation resulting from the variation of junctional length, because both the intensities and the inferred tension are proportional to the inverse of junctional length (Supplementary Fig. 2D). It also yielded the correlation

coefficient as a function of junctional length (Figs. 2i, 6d, f; Supplementary Fig. 2C). We implemented the conditional correlation by sorting 10 junctions of the same length into the same bin. A linear correlation coefficient was computed among these 10 junctions with the same length for Vinculin and E-cadherin intensity (Supplementary Fig. 2C), E-cadherin intensity and inferred shear (Fig. 6d, f), junctional intensity and inferred tension (Fig. 2i). The binning was performed independently for each snapshot to avoid temporal and inter-embryo variations. Finally, we obtained the distribution of the conditional correlation coefficient by combining all the bins across time points and embryos (Figs. 2h, 6b, c, e; Supplementary Fig. 2B).

**Code availability**. This study uses custom codes (MATLAB scripts and ImageJ macros). However, the codes themselves have no bearing on the results, and are only used to facilitate batch processing of the data. The central ideas for the analyses have been described in the Methods section along with Supplementary Figures 5 and 6. Any code requests can be directed to the authors.

## Data availability
The data-sets that support the findings of this study are available from the corresponding authors upon reasonable request.

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

## Acknowledgements

We thank members of Lecuit and Lenne group for discussions throughout the course of this project and for providing a stimulating environment, Claire Chardes (Lenne group) and Claudio Collinet (Lecuit group) for assistance with laser ablation experiments, IBDM imaging facility for assistance with maintenance of the Microscopes, and FlyBase for maintaining curated database and Bloomington fly facility for providing transgenenic flies. G.R.K. was supported by Ph.D. fellowship from the LabEx INFORM (ANR-11-LABX-0054) and of the A*MIDEX project (ANR-11-IDEX-0001-02), funded by the "Investissements d'Avenir French Government program". We also thank NCBS-TIFR for a bridging post-doctoral fellowship for G.R.K., hosted in the laboratory of Satyajit Mayor during the completion of this manuscript. We acknowledge France-BioImaging infrastructure supported by the French National Research Agency (ANR-10-INBS-04-01, «Investments for the future»).

## Author contributions

G.R.K., X.Y., M.M., P.-F.L. and T.L. planned the project. G.R.K. did the experiments, X.Y. performed the mechanical inference studies, J.-M.P. did the constructs. G.R.K. and X.Y. did the analysis. All authors discussed the data. G.R.K., X.Y. and T.L. wrote the manuscript and all authors gave comments on the manuscript.

## Additional information

**Competing interests:** The authors declare no competing interests.

