## [Peer Review File · Nature Communications]

Reviewers' comments:

Reviewer #1 (Remarks to the Author):

This interesting paper explores how correlations between myosin II accumulation and actin fiber recoil (as indicators of tension) influence levels of E-cadherin at cell-cell junctions in *Drosophila*. There are actually two findings in this paper. The first is that E-cadherin/vinculin ratio correlates better with 'inferred tension' than E-cadherin, vinculin, or myosin II levels. Arriving at this conclusion required careful, high-resolution quantitative imaging and co-localization measurements. This result is not entirely surprising, but it provides a useful, relatively robust marker for relating different perturbations to tension on E-cadherin junctions. This useful quantitative measurement tool was then used to infer how different forces on E-cadherin stabilized or destabilized E-cadherin at cell junctions. This resulted in the final very interesting proposal that medial myosin mainly generate forces orthogonal to junctions and this results in E-cadherin accumulation. Conversely, shear forces attributed to junctional myosin correlates with a reduction in E-cadherin at junctions. Thus, the two pools potentially play complementary roles in regulating junctional E-cadherin levels and the stability of cell-cell contacts.

A problem with this manuscript is that it was difficult to follow, primarily because the authors have not defined medial and junctional myosin II in this work. That became clear only after reading through several of their prior papers before finding where this is explicitly defined. Supplemental figure 1A is somewhat helpful, but doesn't clarify why one would pool would generate orthogonal versus transverse forces relative to the junction. They also need to explicitly state in the methods how they quantified the separate pools. The methods section focuses on using segmentation algorithms used to define the junctions based on protein levels, but did not define the medial pool or how it was quantified, including background subtraction. How they quantified polarity should also be better described.

On page 7, they state that they selectively activated the medial myosin II pool without affecting myosin II recruitment to the junctional pool. However, Figure 4C only shows myosin II levels, not the myosin II activity status. They appear to assume that the myosin II accumulation is synonymous with its activation, but it's not obvious why that would be the case. This needs to be clarified.

The title may be stronger than the data support. The correlation of medial myosin II levels (increase or decrease) with Vinc/E-cadherin ratios and E-cadherin levels is compelling. However, the evidence that shear forces disrupt E-cadherin adhesion is weaker, as there are not similar perturbations that increased or decreased shear forces with corresponding changes in E-cadherin levels.

The tissue level forces on vertical and transverse junctions might be expected to differ. Little was said about the mechanical differences and how they might also affect the E-cadherin distribution and its stability at junctions, yet in some of the images, there do appear to be some differences in E-cadherin levels vertical versus transverse junctions.

In general, the statistical analyses appear to be rigorous.

Reviewer #2 (Remarks to the Author):

The *Drosophila* embryonic ectoderm undergoes a well-studied cell rearrangement process based on the remodeling of epithelial junctions. The cells are linked through E-cadherin, and the shrinkage of vertical, but not transverse cell-cell boundaries is driven by myosin II activity. Myosin II is enriched at vertical junctions, but is also present medially underneath the apical cell membrane. In a series of elegant experiments, the authors distinguish between the effects of

junctional and medial myosin II on the stresses exerted on E-cadherin, and on E-cadherin density and distribution. They suggest that respective differences between "tensile" and shear stress can explain the stabilization vs. the dynamic remodeling of cell adhesion. The paper addresses an important topic in an interesting way. However, several main issues remain insufficiently resolved and make it difficult to interpret the experimental findings.

1 Relationship between junctional tension and myosin II levels, Vinc/E-cad ratios (Fig.2B,C). It is shown that myosin II levels and Vinc/E-cad ratios are correlated with the recoil velocity after laser ablation of junctions. Though statistically significant, it seems that a simple linear correlation does not justice to the data shown in Fig.2B,C. Data points are scattered over large areas of characteristic shapes, and drawing regression lines through these "clouds" seems an arbitrary way to extract the relevant information contained in the data. As an example of the extent of data point scattering: at normalized myosin II intensity 1, recoil velocity varies over a 6-fold range, and similarly, at a velocity of 0.2 $\mu\text{m/s}$ myosin density varies over the same range. The same variability is seen in Fig.2C. It seems that the largest part of the tension at junctions is not explained by myosin II intensity. This point should be explicitly addressed.

A possible explanation for the low correlation could be that passive elastic tensions of the actin cortex play a role, and that for this reason myosin II densities in junctions adjacent to the ablated ones are relevant. In the description of the mechanical interference approach in the Methods sections, the authors assume that the junction network is always close to force equilibrium. Is this generally assumed? And is it assumed that myosin II contractility is the only source of tensions? If so, this should be mentioned in the main text, discussed and justified.

2 Related to this: The model for shear stress in Fig.6A and the equation therein should be better explained. Are we looking at a force equilibrium? Are T1, T2 etc. vectors? If not, what about the angles between junctions? How is the equation for the shear stress derived? The two sentences at the end of the Mechanical inference section on p. 14 are definitively not sufficient to allow the reader to critically evaluate the model.

Importantly, junctional myosin is depicted as equally dense on both sides of junction T. How is this assumption justified? Having uneven densities (higher on one side of the boundary, lower on the other) should also generate shear stress, which would not be picked up by the mechanical inference procedure used, however. Perhaps a mosaic expression of differently labeled myosin II could solve the descriptive part of this problem.

3 p.7 first paragraph and Fig.4. Reduced activation of myosin in medial pool decreases Vinc/E-cad ratio: this is not as clear a result as stated. Unfortunately, axes in the diagrams are not to the same scale, and the variation between different treatments is easily overlooked. It seems that the controls for the medial myosin inhibition experiment are unusual. Junctional myosin is unusually low (Fig.4C'') compared to the other experiments, and the control Vinc/E-cad ratio is almost twice that of the other controls (Fig.4D'). The error bar is also largest in this sample, and if this control set shows indeed an artificially high ratio (by random fluctuation?), reduction of the ratio by RhoGEF RNAi is questionable: the Vinc/E-cad ratio is numerically the same as after Rok inhibition (Fig.4B').

4 p.7 first paragraph and Supplementary Fig. 5B. Reduction of junctional tension by medial myosin inhibition: not only is the recoil velocity not diminished, as implied in the text, but it is significantly increased. This suggests that medial myosin normally reduces junctional tensions, which complicates the interpretation of the data, probably including those derived from the mechanical inference procedure.

5 p.9 line 3. Vinc/E-cad ratio being greater at brighter E-cad pixels. It should be explained how this fits to the E-cad overexpression results, where the ratio is reduced by additional E-cad.

6 p.10 last paragraph. The speculation on the functions of different types of stress on E-cadherin junctions is appealing, but it is not strongly supported by the data shown. The argument is made to rely on the turnover of cadherin bonds: "on average the density of complexes could remain constant but the turnover of homophilic bonds would be increased." No data on the turnover are provided; in fact, a reduction of E-cadherin intensity at vertical boundaries, expressed in the form of correlation coefficients, is taken as evidence for the proposed mechanism. Also, showing direct E-cad intensity measurements would help the reader to see at least intuitively whether the intensity differences (an estimated -0.1 difference in the correlation coefficient, Fig.6C) could be significant mechanistically.

Overall, the manuscript in its present form contains many loose ends which severely affect the possible interpretations of the data.

Reviewer #3 (Remarks to the Author):

In this manuscript, the authors combine experimental and computational methods to study the role of medial and junctional MyoII in regulating adhesion forces between epithelial cells. They first perform experiments to establish the ratio of Vinc/E-Cad as a readout of force. Then, through laser cuts they find that Vinc/E-cad, as well as MyoII correlate with tension. They use mechanical inference to calculate junctional tensions starting from the Vinc/E-Cad ratio of tens of thousands of junctions. They look at the different contributions of medial and junctional Myosin-II to the organization of adhesion complexes and find evidence that shear stress regulates E-Cadherin levels at junctions.

This is a solid study relying on fluorescence intensity measurements and computational estimation of junctional tension. The central aspect of this study is to establish the Vinc/E-cad ratio as a readout to adequately estimate the effects of MyoII activity on junctional remodeling. The approach is interesting but there are some doubts left, particularly about the suitability of the methods used for validation.

In the following a list of concerns and points that need to be addressed:

- Referring to Vinculin, or Vinc/Ecad as a "force sensor" is misleading. Reading the abstract gives the impression that the authors generated something like the previously published Vinculin FRET sensor (<https://dx.doi.org/10.1038%2Fnature09198>). Using terminology such as "ratiometric readout" or other would avoid such confusion.
- What is the advantage of using Vinc/Ecad fluorescence ratios rather than junctional MyoII signal intensities to estimate tension? After all, the authors base all their conclusions and the validation of the Vinc/Ecad fluorescence ratios as adequate readout for tension, on the assumption that MyoII fluorescence changes fully correlate with changes in MyoII force generating activity, despite a borderline p-value in Fig. 2B (btw also in Fig. 2C and suppl. Fig. 3A p-values are borderline).
- Vinculin seems to have highest signal intensity at cell vertices (e.g. Fig. 1A) which is also where it best correlates with the E-cadherin signal. Yet, this signal is excluded from measurement. It seems problematic to assume that this significant protein pool is irrelevant with respect to the generation of junctional tension or to handling shear.
- Under differing conditions junctions seem to become wiggly, in particular the transverse junctions (e.g. Fig. 5B). How is this accommodated when segmenting and measuring? Are these junctions lacking tension because they are squeezed together by medial MyoII activity? Laser cuts would be revealing.
- Mechanical inference was set up to be used for tissues near mechanical equilibrium and should be used with care in systems with dynamic Myosin behavior. This raises doubts about its suitability for computing tension in the actively stretching tissue used in this study. A possible way to validate would be to apply inference analysis to the last frames taken before laser ablation of junctions. Do recoil velocity and predicted tension correlate well?
- In their analysis the authors ignore the timescales of force generation and junction remodeling. On pages 4, 8, and 9 the authors refer to shrinking junctions but without time resolved analysis

this is merely an assumption. Using mechanical inference on consecutive time frames of a movie sequence would allow correlating measurement and shrinking behavior of junctions and provide further validation of the method.

- Not enough information is provided to understand how the inference method is implemented. The authors should elaborate on how they used what was published in Noll et al. 2017, the main reference given in the methods section (page 14). With the current information independent reproduction is not possible.
- The authors do no comment on why the particular correlation measures were chosen (e.g. Spearman correlation in Fig. 2B, 2C and Pearson correlation computed in local correlation in Fig. 2H-J). Also, why is local correlation needed given that tensions are relative and junctional intensity is normalized?
- It is worrisome when control values of the same measurements in different experiments differ significantly:
 - a) Quantification of junctional MyoII in Fig. 1D and 4A. In particular, the levels of the Rok Inhibition experiment seem to differ significantly.
 - b) The control values of Vinc/E-cad ratios vary considerably between Fig. 4D' and Fig. 4B', 4F'.
 - c) Similarly, the control levels of E-cad in Fig. 5A'-B' differ significantly.How do the authors explain this variation?

Minor comments:

P. 6: the comment in square brackets was probably not meant to get to reviewers

- Missing reference, page 10, last paragraph: "this condition is observed when adhesion is mildly compromised using partial knock-down of E-cadherin or α -catenin in the mesoderm and endoderm where medial Myosin-II accumulates at very high level"
- Page 7, 8, 9: It is not clear what is meant with "uniform" loading

We thank the reviewers for their constructive and thoughtful comments. We have addressed all of their concerns which we detail below.

Reviewer 1:

This interesting paper explores how correlations between myosin II accumulation and actin fiber recoil (as indicators of tension) influence levels of E-cadherin at cell-cell junctions in *Drosophila*. There are actually two findings in this paper. The first is that E-cadherin/vinculin ratio correlates better with 'inferred tension' than E-cadherin, vinculin, or myosin II levels. Arriving at this conclusion required careful, high-resolution quantitative imaging and co-localization measurements. This result is not entirely surprising, but it provides a useful, relatively robust marker for relating different perturbations to tension on E-cadherin junctions. This useful quantitative measurement tool was then used to infer how different forces on E-cadherin stabilized or destabilized E-cadherin at cell junctions. This resulted in the final very interesting proposal that medial myosin mainly generate forces orthogonal to junctions and this results in E-cadherin accumulation.

Conversely, shear forces attributed to junctional myosin correlates with a reduction in E-cadherin at junctions. Thus, the two pools potentially play complementary roles in regulating junctional E-cadherin levels and the stability of cell-cell contacts.

1) A problem with this manuscript is that it was difficult to follow, primarily because the authors have not defined medial and junctional myosin II in this work. That became clear only after reading through several of their prior papers before finding where this is explicitly defined. Supplemental figure 1A is somewhat helpful, but doesn't clarify why one would pool would generate orthogonal versus transverse forces relative to the junction.

Response: In supplementary Figure 1, we have now added a schematic showing the apical view of the cells to help visualize the two pools of Myosin-II. The main text (introduction) now has a more detailed description of the two pools and how they exert orthogonal or tangential forces. Laser ablation experiments targeting the medial actomyosin network (Levayer et al 2013 and Collinet et al 2015) or the junctional network (Rauzi et al 2008) support this view. When junctional Myosin-II is ablated, junctions relax in a tangential direction, whereas medial ablation causes relaxation of the junction in a direction orthogonal to the junction.

2) They also need to explicitly state in the methods how they quantified the separate pools. The methods section focuses on using segmentation algorithms used to define the junctions based on protein levels, but did not define the medial pool or how it was quantified, including background subtraction. How they quantified polarity should also be better described.

Response: We have rewritten the method sections that describe the intensity measurements workflow in greater details and have also added a supplementary figure to help visualize the workflow.

D Follow-up from **C'**: Segregate 'line density' values (ρ) for junctions (j) based on their angle relative to the AP axis for each embryo (E). Using that, calculate 'averaged line density' (ALD) and 'planar cell polarity' (PCP) for each of 'N' embryos. These values are used to calculate 'mean junctional intensity' and 'amplitude of polarity'. The same values are used to estimate the related SEM values.

Angle bins	(AP)	15-30	30-45	45-60	60-75	(DV)
	00-15					75-90
	$\rho(j_a)$	$\rho(j_b)$	$\rho(j_c)$	$\rho(j_d)$	$\rho(j_e)$	$\rho(j_f)$
	$\rho(j_g)$	$\rho(j_h)$	$\rho(j_i)$	$\rho(j_j)$	$\rho(j_k)$	$\rho(j_l)$
	$\rho(j_m)$	$\rho(j_n)$	$\rho(j_o)$	$\rho(j_p)$	$\rho(j_q)$	$\rho(j_r)$
	$\rho(j_s)$	$\rho(j_t)$.	$\rho(j_u)$	$\rho(j_v)$	$\rho(j_w)$
	$\rho(j_x)$.	.	$\rho(j_y)$.	$\rho(j_z)$

Averages for each categories	$AVG_{(00-15)}$	$AVG_{(15-30)}$	$AVG_{(30-45)}$	$AVG_{(45-60)}$	$AVG_{(60-75)}$	$AVG_{(75-90)}$

$$ALD(E_1) = (AVG_{(00-15)} + AVG_{(15-30)} + \dots + AVG_{(75-90)}) / 6 \rightarrow \text{mean junctional intensity} = (ALD(E_1) + ALD(E_2) + \dots + ALD(E_N)) / N$$

$$PCP_{DV/AP}(E_1) = AVG_{(75-90)} / AVG_{(00-15)} \rightarrow \text{amplitude of polarity}_{DV/AP} = (PCP_{DV/AP}(E_1) + PCP_{DV/AP}(E_2) + \dots + PCP_{DV/AP}(E_N)) / N$$

E Follow-up from **C''**: Pool 'area density' values (R) for 'n' number of cells per image. Using that, calculate 'averaged area density' (AAD) for each of 'N' embryos (E). These values are used to calculate 'mean medial intensity'. The same values are used to estimate the related SEM values.

$$AAD(E_1) = (R_1 + R_2 + \dots + R_n) / n \rightarrow \text{mean medial intensity} = (AAD(E_1) + AAD(E_2) + \dots + AAD(E_n)) / n$$

3) On page 7, they state that they selectively activated the medial myosin II pool without affecting myosin II recruitment to the junctional pool. However, Figure 4C only shows myosin II levels, not the myosin II activity status. They appear to assume that the myosin II accumulation is synonymous with its activation, but it's not obvious why that would be the case. This needs to be clarified.

Response: We indeed assume that the Myosin-II accumulation/ localization reflects its activation, since lack of phosphorylation (activation) prevents Myosin-II recruitment, as seen in Rok inhibition experiments (Fig. 1D, Fig. 4A. Also, see for instance Munjal et al 2015). Moreover, phosphomimetic Myosin-II is recruited to F-actin meshwork in spite of Rok inhibition (Munjal et al. 2015). We have talked about how medial Myosin-II can be specifically tuned based on Kerridge et al. (2016) where we reported that $\text{G}\alpha_{12/13}$ and RhoGEF2 control specifically medial MyoII activation. We have explicitly mentioned this in the ‘introduction’.

“Based on previous studies, Myosin-II activation by phosphorylation of its regulatory light chain can be directly inferred from its recruitment (Munjal et al. 2015). Thus, we used changes in Myosin-II recruitment as a proxy for changes in its activation and for the changes in the generation of tensile forces themselves. Myosin-II phosphorylation depends on the kinase Rok, which is activated by the small GTPase Rho1. Medial activation of Rho1 depends on $\text{G}\alpha_{12/13}$ (also called *Concertina*) and its molecular effector, the GEF RhoGEF2 (Kerridge et al. 2016). Thus $\text{G}\alpha_{12/13}$ and RhoGEF2 control medial apical actomyosin tension by specifically regulating apical actomyosin recruitment.”

4) The title may be stronger than the data support. The correlation of medial myosin II levels (increase or decrease) with Vinc/E-cadherin ratios and E-cadherin levels is compelling. However, the evidence that shear forces disrupt E-cadherin adhesion is weaker, as there are not similar perturbations that increased or decreased shear forces with corresponding changes in E-cadherin levels.

Response: This is a fair point and we now provide direct evidence for this. Laser ablation of junctions abolishes tension in targeted junction and therefore reduces or increases shear in adjacent junction depending on which junctions are targeted (see Fig below). Therefore, we ablated junctions and monitored changes in E-cadherin density in adjacent junctions which experience increase in shear stress over a time interval of 20 seconds after ablation. As discussed in the schematics shown in Figure 6G, an ablation of T_2 or T_4 increases shear following the definition of shear stress, provided $(T_1+T_3)>(T_2+T_4)$. An increase of shear stress by ablation leads to a reduction in mean E-cadherin intensity as compared to control junctions neighboring to a non-ablated junction. We have also shown that the decrease in mean E-cadherin intensity upon increased shear is NOT due to a dilution effect from the elongation of junction, because the junction actually shrinks. Thus, this decrease in mean E-cadherin intensity is a result of the decrease in the number of E-cadherin molecules. These new data are now presented in Figure 6H and page 8-9.

5) The tissue level forces on vertical and transverse junctions might be expected to differ. Little was said about the mechanical differences and how they might also affect the E-cadherin distribution and its stability at junctions, yet in some of the images, there do appear to be some differences in E-cadherin levels vertical versus transverse junctions.

Response: Tissue scale forces do indeed decrease in the ectoderm from the posterior towards the anterior (Collinet et al. 2015). This tissue scale tension is due to posterior endoderm invagination. This study used laser ablation to assess tissue scale tension anisotropy and found that the tension was more isotropic towards the posterior because tissue scale tension there is commensurate with local Myosin-II dependent tension in vertical junctions (Figure 7c, d, e, j in Collinet et al. 2015). In the experiments presented in the submitted manuscript, we have performed imaging in the anterior half of the ectoderm where tissue scale forces are negligible (as shown earlier in Collinet et al 2015). Hence, tension is anisotropic and contributed only by local Myosin-II. We have not extended our analysis to the posterior because of the rapid movement of the cells towards the posterior and the curvature of the embryo which complicates the analysis. Now we have given a more detailed description of tension anisotropy in the cells in the 'introduction' (page 3).

In general, the statistical analyses appear to be rigorous.

Thank you.

Reviewer #2

The *Drosophila* embryonic ectoderm undergoes a well-studied cell rearrangement process based on the remodeling of epithelial junctions. The cells are linked through E-cadherin, and the shrinkage of vertical, but not transverse cell-cell boundaries is driven by myosin II activity. Myosin II is enriched at vertical junctions, but is also present medially underneath the apical cell

membrane. In a series of elegant experiments, the authors distinguish between the effects of junctional and medial myosin II on the stresses exerted on E-cadherin, and on E-cadherin density and distribution. They suggest that respective differences between “tensile” and shear stress can explain the stabilization vs. the dynamic remodeling of cell adhesion. The paper addresses an important topic in an interesting way. However, several main issues remain insufficiently resolved and make it difficult to interpret the experimental findings.

1) Relationship between junctional tension and myosin II levels, Vinc/E-cad ratios (Fig.2B,C). It is shown that myosin II levels and Vinc/E-cad ratios are correlated with the recoil velocity after laser ablation of junctions. Though statistically significant, it seems that a simple linear correlation does not justice to the data shown in Fig.2B,C. Data points are scattered over large areas of characteristic shapes, and drawing regression lines through these “clouds” seems an arbitrary way to extract the relevant information contained in the data.

Response: This is a fair point and we are happy to clarify. First, we may have not been clear enough (though this was stated in the Figure legends and in Methods), we are using ‘Spearman correlation’ to get the correlation coefficient and to check the statistical significance. Importantly, Spearman correlation does not assume a linear relationship between the variables. So we do not draw regression lines, as this would be an arbitrary way to extract relevant information contained in the data.

As an example of the extent of data point scattering: at normalized myosin II intensity 1, recoil velocity varies over a 6-fold range, and similarly, at a velocity of 0.2 $\mu\text{m/s}$ myosin density varies over the same range. The same variability is seen in Fig.2C. It seems that the largest part of the tension at junctions is not explained by myosin II intensity. This point should be explicitly addressed.

Response: We thank the reviewer for prompting a further investigation of the data scattering. Analysis of recoil velocities requires pooling together data from different embryos at different developmental times, which introduces scattering due to sample variations and changes in imaging conditions. To overcome this limitation of laser ablation, we have used mechanical inference as an independent method to estimate tension distribution, which suffers much less from the aforementioned variabilities and generates enough data for statistics within a single image for one embryo at one developmental time point. We have performed comparative analysis to show that correlation coefficient between Vinc/ECad and inferred tension within a single image is about 0.5 as compared to 0.25 for inter-embryo correlations.

G correlation with inferred tension
(single image)

We also showed that Vinc/ECad displays a comparable correlation (in the range of 0.25) with recoil velocity as with inferred tension in inter-embryo correlations once normalized to reduce as much as possible inter-embryo variability (see below). The scattering of the data is likely due to inter-embryo differences and the relatively small sample size ($N=25-60$). This indicates that inferred tension is at least as good a tension estimate as recoil velocity. Given that mechanical inference is unaffected by inter-embryo variability and produces much more data than laser ablation, we consistently estimate tensions using mechanical inference throughout the manuscript.

E correlation with recoil velocity
(across embryos)

F correlation with inferred tension
(across embryos)

A possible explanation for the low correlation could be that passive elastic tensions of the actin cortex play a role, and that for this reason myosin II densities in junctions adjacent to the ablated ones are relevant.

Response: This is an interesting possibility. We have tested this by measuring the recoil velocities in the neighboring junctions. In general, we don't see a strong recoil due to ablations in the neighboring junctions so we think the contribution of adjacent junctions is minor/negligible. We have not included this analysis, as its conclusions don't affect the interpretation of the data presented in the manuscript. (D=dorsal, V=ventral, A=anterior, P=posterior)

In the description of the mechanical interference approach in the Methods sections, the authors assume that the junction network is always close to force equilibrium. Is this generally assumed?

Response: The assumption of force equilibrium is supported by the observation that the recoil velocity of junctions after laser ablation is much larger than the velocity of the cell and vertices during development, indicating that junctional tensions are much larger than the unbalanced force at cell vertices. Therefore, force balance at cell vertices is a good approximation. We have now explicitly stated this in the 'methods' section.

And is it assumed that myosin II contractility is the only source of tensions? If so, this should be mentioned in the main text, discussed and justified.

Response: We have evidence that Myosin-II contractility is the major source of tension from previous laser ablation experiments (Rauzi et al, 2008, Collinet et al, 2015) and optical tweezers experiments (Bambardekar et al 2015).

2 Related to this: The model for shear stress in Fig.6A and the equation therein should be better explained. Are we looking at a force equilibrium? Are T1, T2 etc. vectors? If not, what about the

angles between junctions? How is the equation for the shear stress derived? The two sentences at the end of the Mechanical inference section on p. 14 are definitively not sufficient to allow the reader to critically evaluate the model.

Response: We assumed force balance at each vertex as in mechanical inference. T_1, T_2 , etc. are magnitudes of tensions as inferred from mechanical inference. Shear stress is generated as a result of the transfer of tension from one side of the junction to the other as derived by Chiou et al. (2012). We have now explicitly reproduced the derivation of the shear stress in Methods section. We also show the derivation below for clarity:

Shear stress on the E-cadherin clusters was obtained from a microscopic model of the junction (Fig. 6A) (Chiou, Hufnagel, and Shraiman 2012). As illustrated in Figure 6A, tension of the central junction is decomposed into cortical tensions at a-cell side and b-cell side of the junction: $T = \theta_a(x) + \theta_b(x)$, where x is the coordinate along the junction. While T is constant along the junction, cortical tensions $\theta_a(x)$ and $\theta_b(x)$ can vary along the junction in opposing gradients (red lines in Figure 6XX) as a result of the transfer of tension from one side of the junction to the other. This transfer of tension generates shear stress on E-Cadherin dimers. Therefore, shear stress at any given point along the junction is defined as the gradient of cortical tension $\tau(x) = \partial_x \theta_a(x) = -\partial_x \theta_b(x)$. The average shear stress along the junction is $\tau = \frac{1}{l} \int_0^l \tau(x) dx = \frac{\theta_a(l) - \theta_a(0)}{l} = \frac{\theta_b(0) - \theta_b(l)}{l}$. To relate shear stress to junctional tension as obtained from mechanical inference, we assume that cortical tensions are single-valued, i.e. continuous, at vertices, from which we get the relation:

$$\begin{aligned} T &= \theta_a(0) + \theta_b(0) & T_1 &= \theta_b(0) + \theta_d(0) & T_2 &= \theta_a(0) + \theta_d(0) \\ T &= \theta_a(l) + \theta_b(l) & T_3 &= \theta_a(l) + \theta_c(l) & T_4 &= \theta_b(l) + \theta_c(l) \end{aligned}$$

We solve equations above to get cortical tensions in terms of junctional tensions and substitute to the equation for shear stress to obtain the final expression of shear stress:

$$\tau = \frac{1}{2l} |(T_1 + T_3) - (T_2 + T_4)|$$

where l is the length of the junction.

Importantly, junctional myosin is depicted as equally dense on both sides of junction T. How is this assumption justified? Having uneven densities (higher on one side of the boundary, lower on the other) should also generate shear stress, which would not be picked up by the mechanical inference procedure used.

Response: As illustrated in the shear stress model above, shear is generated as a result of the transfer of cortical tension across the junction, which requires uneven cortical tensions at two sides of the junction. An uneven density of Myosin-II could certainly be a source of uneven cortical tensions. Further, we observed an asymmetric distribution of neighboring junctional tension (i.e. $T_1+T_3 > T_2+T_4$), thus as an alternative, we assume an asymmetric distribution of neighboring junctional tensions to generate uneven cortical tensions as depicted in Figure 6A in the main text.

However, perhaps a mosaic expression of differently labeled myosin II could solve the descriptive part of this problem.

Response: Mosaic expression would probably be the best way to test the asymmetry of Myosin-II. Though, it is not possible in the embryos as it is post-mitotic and clonal analysis is not possible at this stage. Perhaps, super-resolution imaging can resolve the cortical Myosin-II from neighboring cells, though that would be out of the scope of current work.

3 p.7 first paragraph and Fig.4. Reduced activation of myosin in medial pool decreases Vinc/E-cad ratio: this is not as clear a result as stated. Unfortunately, axes in the diagrams are not to the same scale, and the variation between different treatments is easily overlooked. It seems that the controls for the medial myosin inhibition experiment are unusual. Junctional myosin is unusually low (Fig.4C'') compared to the other experiments, and the control Vinc/E-cad ratio is almost twice that of the other controls (Fig.4D'). The error bar is also largest in this sample, and if this control set shows indeed an artificially high ratio (by random fluctuation?), reduction of the ratio by RhoGEF RNAi is questionable: the Vinc/E-cad ratio is numerically the same as after Rok inhibition (Fig.4B').

Response: The variation across different treatments can be easily explained by the experimental conditions. Depending on the experiments, the embryos may have different maternal and zygotic genotypes (detailed in, Methods: Fly lines and genetics), thus have a different proportion of tagged versus untagged protein pool. This will change the absolute values but not the relative changes. The entire protein pool is tagged in the case of the Rok inhibition experiment, thus the intensity values are highest. Entire maternal, but only half zygotic protein pool is tagged in Gα12/13 overexpression experiment (an F1 genetic cross), thus the intensity values are lower than the Rok inhibition experiment. Half of maternal as well as

zygotic protein pool is tagged in RhoGEF2 RNAi experiment (an F2 genetic cross), thus the intensity values are about half of Rok inhibition experiment. Thus the proportion of fluorescently tagged protein pool is different; proportion being highest in Rok inhibition experiments and lowest in RhoGEF2-RNAi.

Further, this leads us to adjust/optimize imaging conditions for each treatment for simultaneous acquisition of two proteins; avoiding bleed through from GFP to mCherry channel, avoiding photo-bleaching in either channels and trying to get best signal possible. mCherry seems to bleach faster than GFP, thus imaging conditions need to be altered more for mCherry tagged protein as compared to GFP tagged ones. This makes it particularly difficult to compare Myosin-II intensities across treatments, as the presented intensity quantifications are mostly from mCherry tagged construct. Similar problem also exists for Vinculin-mCherry, and as a consequence also for Vinc/E-cad ratio. Since E-cadherin is GFP tagged, the imaging conditions could be kept most similar across treatments. This allows one to compare the E-cadherin junctional intensities for controls in Figure 1F', 5A' and 5B' and to gauge the variability due to changes in the proportion of untagged protein pool.

Naturally, each treatment is accompanied with respective controls, which have the same proportion of untagged protein pool and are imaged under identical conditions. Thus, we can confidently attribute the changes between treatments and respective controls to the perturbation itself. Also, the quantifications are extremely robust when we present normalized intensities (Amplitude of polarity) as compared to mean intensities. E.g. compare the 'Amplitude of polarity' quantifications for junctional Myosin-II (controls in Figure 4A'', 4C'' and 4E'') as well as Vinc/E-cad ratio (controls in Figure 4B', 4D' and 4F'). Such normalization readily accounts for intensity changes due to changes in proportion of untagged proteins, as both numerator and denominator change proportionally. We have now tried to make this more explicit in the figure legends and have made appropriate connections with relevant 'methods' sub-sections.

4 p.7 first paragraph and Supplementary Fig. 5B. Reduction of junctional tension by medial myosin inhibition: not only is the recoil velocity not diminished, as implied in the text, but it is significantly increased. This suggests that medial myosin normally reduces junctional tensions, which complicates the interpretation of the data, probably including those derived from the mechanical inference procedure.

Response: The recoil velocity is a function of local tension over viscosity. One possibility is that medial actomyosin meshwork is providing some sort of viscous dampening to reduce the junctional recoil in WT embryos. In RhoGEF2 RNAi, the lower actin and medial Myosin-II concentration, due to reduced Rho1 activity, could reduce viscous resistance during junction relaxation and thereby increase relaxation velocity. It will be very interesting to explore this possibility, though we think this is beyond the scope of this manuscript.

5 p.9 line 3. Vinc/E-cad ratio being greater at brighter E-cad pixels. It should be explained how this fits to the E-cad overexpression results, where the ratio is reduced by additional E-cad.

Response: E-cadherin overexpression decreases the baseline recruitment of Vinculin to E-cadherin clusters by reducing the average load. On top of this baseline, Vinc/E-cad ratio increases with E-cadherin cluster size. This data indicates that larger E-cadherin clusters bear larger load, which is consistent with an independent study (Biswas et al. 2016). We have now explained this point in the appropriate results section.

6 p.10 last paragraph. The speculation on the functions of different types of stress on E-cadherin junctions is appealing, but it is not strongly supported by the data shown. The argument is made to rely on the turnover of cadherin bonds: “on average the density of complexes could remain constant but the turnover of homophilic bonds would be increased.” No data on the turnover are provided; in fact, a reduction of E-cadherin intensity at vertical boundaries, expressed in the form of correlation coefficients, is taken as evidence for the proposed mechanism. Also, showing direct E-cad intensity measurements would help the reader to see at least intuitively whether the intensity differences (an estimated -0.1 difference in the correlation coefficient, Fig.6C) could be significant mechanistically.

Response: This is a fair point and we now provide direct evidence for this. Laser ablation of junctions abolishes tension in targeted junction and therefore reduces or increases shear in adjacent junction depending on which junctions are targeted (see Fig below). Therefore, we ablated junctions and monitored changes in E-cadherin density in adjacent junctions which experience increase in shear stress over a time interval of 20 seconds after ablation. As discussed in the schematics shown in Figure 6G, an ablation of T_2 or T_4 increases shear following the definition of shear stress, provided $(T_1+T_3) > (T_2+T_4)$. An increase of shear stress by ablation leads to a reduction in mean E-cadherin intensity as compared to control junctions neighboring to a non-ablated junction. We have also shown that the decrease in mean E-cadherin intensity upon increased shear is NOT due to a dilution effect from the elongation of junction, because the junction actually shrinks. Thus this decrease in mean E-cadherin intensity is a result of the decrease in the number of E-cadherin molecules. These new data are now presented in Figure 6H and page 8-9.

The comment about turnover of homophilic bonds is a hypothesis presented in the discussion. Addressing this point is out of the scope for current study, given the spatio-temporal resolution limits.

Reviewer #3

In this manuscript, the authors combine experimental and computational methods to study the role of medial and junctional MyoII in regulating adhesion forces between epithelial cells. They first perform experiments to establish the ratio of Vinc/E-cad as a readout of force. Then, through laser cuts they find that Vinc/E-cad, as well as MyoII correlate with tension. They use mechanical inference to calculate junctional tensions starting from the Vinc/E-cad ratio of tens of thousands of junctions. They look at the different contributions of medial and junctional Myosin-II to the organization of adhesion complexes and find evidence that shear stress regulates E-cadherin levels at junctions.

This is a solid study relying on fluorescence intensity measurements and computational estimation of junctional tension. The central aspect of this study is to establish the Vinc/E-cad ratio as a readout to adequately estimate the effects of MyoII activity on junctional remodeling. The approach is interesting but there are some doubts left, particularly about the suitability of the methods used for validation.

In the following a list of concerns and points that need to be addressed:

- Referring to Vinculin, or Vinc/Ecad as a "force sensor" is misleading. Reading the abstract gives the impression that the authors generated something like the previously published Vinculin FRET sensor (<https://dx.doi.org/10.1038%2Fnature09198>). Using terminology such as "ratiometric readout" or other would avoid such confusion.

Response: Thank you for the suggestion. We have now included this terminology in the manuscript.

- What is the advantage of using Vinc/Ecad fluorescence ratios rather than junctional MyoII signal intensities to estimate tension?

Response: There are two advantages: (1) Vinc/ECad is a dimensionless ratiometric readout of forces, which is independent of intensity fluctuations caused by variations in junctional lengths, something that can affect Myosin-II intensity estimations. It therefore provides a more robust measurement of tension compared to Myosin-II intensity as shown by a higher correlation coefficient with inferred tension (Fig. 2H, I). (2) We have also tried to establish Vinc/E-cad ratio as an estimate for the force “experienced” by (or acting upon) E-cadherin. This is not synonymous to what is “generated” by Myosin-II across a whole junction. Load on E-cadherin is a quantity of importance for mechanotransduction and the idea is to find an observable to probe that. We have now emphasized these advantages in the ‘discussion’ section of the manuscript (page 9).

After all, the authors base all their conclusions and the validation of the Vinc/Ecad fluorescence ratios as adequate readout for tension, on the assumption that MyoII fluorescence changes fully correlate with changes in MyoII force generating activity, despite a borderline p-value in Fig. 2B (btw also in Fig. 2C and suppl. Fig. 3A p-values are borderline).

Response: Analysis of recoil velocities requires pooling together data from different embryos at different developmental times, which introduces scattering due to sample variations and changes in imaging conditions. To overcome this limitation of laser ablation, we have used mechanical inference as an independent method to estimate tension distribution, which suffers much less from the aforementioned variabilities and generates enough data for statistics within a single image for one embryo at one developmental time point. We have performed comparative analysis to show that correlation coefficient between Vinc/ECad and inferred tension within a single image is about 0.5 as compared to 0.25 for inter-embryo correlations.

G correlation with inferred tension
(single image)

We also showed that Vinc/ECad displays a comparable correlation (in the range of 0.25) with recoil velocity as with inferred tension in inter-embryo correlations once normalized to reduce as much as possible inter-embryo variability (see below). The scattering of the data is likely due to inter-embryo differences and the relatively small sample size ($N=25-60$). This indicates that inferred tension is at least as good a force estimate as recoil velocity. Given that mechanical inference is unaffected by inter-embryo variability and produces much more data than laser ablation, we consistently estimate tensions using mechanical inference throughout the manuscript.

F correlation with recoil velocity
(across embryos)

F correlation with inferred tension
(across embryos)

- Vinculin seems to have highest signal intensity at cell vertices (e.g. Fig. 1A) which is also where it best correlates with the E-cadherin signal. Yet, this signal is excluded from measurement. It

seems problematic to assume that this significant protein pool is irrelevant with respect to the generation of junctional tension or to handling shear.

Response: We present below the correlation between inferred tension and Vinc/ECad at vertices. The inferred tension at a cell vertex is the average of the three junctional tensions meeting at the vertex. While there is a consistent positive correlation, the correlation coefficient (0.12) is much smaller compared to junctional Vinc/ECad (~0.6). This could suggest that junctional tensions are not the most relevant forces at vertices, or that the mechanosensitivity of E-cadherin complexes is different at the vertex due to its unique geometry. As shear is averaged over the junction, it is unclear how to define shear at a vertex. Also, the fact that we see strong E-cadherin signals could be due to the fact that there are three apposed membranes meeting at each vertex which we cannot yet resolve separately. We have made comments in the discussion section about this, and refer to a recent publication (Vanderleest et al. 2018), which has specifically looked at E-cadherin dynamics at the vertices.

- Under differing conditions junctions seem to become wiggly, in particular the transverse junctions (e.g. Fig. 5B). How is this accommodated when segmenting and measuring?

Response: Segmentation pipeline traces the contour directly to get a curvilinear segmentation. The segmented junction itself is then processed as usual.

Are these junctions lacking tension because they are squeezed together by medial MyoII activity? Laser cuts would be revealing.

Response: It is not clear whether the junctions are lacking tension or whether they are squeezed. Though, it is hard to imagine why transverse junctions would be squeezed more than vertical junctions, given that the tissue extension is responsible for stretching the transverse junctions (Collinet et al. 2015). The junctions look wiggly, as medial Myosin-II might be tugging on E-cadherin clusters and deforms/bends the junctions. This effect will be exaggerated due to an increase in medial Myosin-II as well as an increase in E-cadherin levels themselves. We can see a few wiggly junctions even in WT embryos and we don't see them in RhoGEF2-RNAi embryos where medial MyoII is strongly reduced. So, we strongly believe this to be an effect of

medial Myosin-II. Previous studies from the lab have indeed demonstrated that the bent junctions straighten when we make laser cuts in the medial pool.

- Mechanical inference was set up to be used for tissues near mechanical equilibrium and should be used with care in systems with dynamic Myosin behavior. This raises doubts about its suitability for computing tension in the actively stretching tissue used in this study. A possible way to validate would be to apply inference analysis to the last frames taken before laser ablation of junctions. Do recoil velocity and predicted tension correlate well?

Response: We have performed mechanical inference to the last frame before laser ablation. The inferred tension displays a similar positive correlation with recoil velocity as compared to junctional Myosin-II intensity and Vinc/E-cad ratio. Note that correlation measurements between recoil velocity and inferred tension requires to gather data from different embryos; as the inferred tension is known within a prefactor, this might lead to an underestimated correlation. We have now added this data to Supplementary Figure 3C.

Another justification for mechanical equilibrium is that the velocity of the cell and of vertices during development is much smaller than the recoil velocity after ablation, indicating that the unbalanced tensions at vertices are much smaller than junctional tensions.

C

- In their analysis the authors ignore the timescales of force generation and junction remodeling. On pages 4, 8, and 9 the authors refer to shrinking junctions but without time resolved analysis this is merely an assumption.

Response: We have now made corrections in the manuscript to use “shrinking junctions” only in the sections where we present time resolved analysis. In all other cases, we are using the term “vertical junctions” of which “shrinking junctions” is a sub-category.

Using mechanical inference on consecutive time frames of a movie sequence would allow correlating measurement and shrinking behavior of junctions and provide further validation of the method.

Response: Below we are presenting additional analysis along these lines to check if inferred tension in a given frame correlates with junction behavior (shrink or not) in subsequent frames. We first correlated the inferred tension with junction shrinkage rate as computed using two subsequent frames 30 seconds apart. If the junction shrinks under tension, we expect to see a strong negative correlation between inferred tension and shrinkage rate. The data shows that the correlation is low. This is consistent with the hypothesis of force balance at vertices in mechanical inference, because whatever the magnitude of the tension is, it will be balanced by neighboring tensions, hence will not contribute to junctional shrinkage directly.

- Not enough information is provided to understand how the inference method is implemented. The authors should elaborate on how they used what was published in Noll et al. 2017, the main reference given in the methods section (page 14). With the current information independent reproduction is not possible.

Response: We inferred junctional tensions by fitting a tension triangulation network to the segmented cell network. We have now elaborated on the implementation of mechanical inference in the methods section. We also added a schematic plot to Supplementary Figure 3A to illustrate the method.

- The authors do no comment on why the particular correlation measures were chosen (e.g. Spearman correlation in Fig. 2B, 2C and Pearson correlation computed in local correlation in Fig. 2H-J). Also, why is local correlation needed given that tensions are relative and junctional intensity is normalized?

Response: Spearman correlation' looks for monotonic relationships, thus allowing relationships to be non-linear as well. 'Pearson correlation', in contrast, looks for linear relationships. In our analyses, the differences between 'Spearman correlation coefficients' and 'Pearson correlation coefficients' are minimal, if any, indicating that most relationships are linear. Though, we have used 'spearman correlation' when the number of data points is small <100, where an assumption of linear relationship might not be justified due to fewer data points. We have used 'Pearson correlation' in all other cases. Further, we have not normalized junctional intensities in

the local correlation analysis, as normalization does not get rid of the spatial variations of junctional intensities over the field of view due to, e.g. laser power fluctuations or tissue height fluctuations. By construction, local correlation is not affected by such fluctuations as laser power and tissue height are relatively constant over the scale of a single cell.

- It is worrisome when control values of the same measurements in different experiments differ significantly:

a) Quantification of junctional MyoII in Fig. 1D and 4A. In particular, the levels of the Rok Inhibition experiment seem to differ significantly.

Response: Figure 1D and 4A have respectively, GFP tagged and mCherry tagged Myosin-II and hence the intensities should not be compared directly. The imaging conditions were quite different, as GFP is brighter than mCherry and consequently MyoII-GFP is brighter than MyoII-mCherry. The imaging conditions are also differently optimized as MyoII-GFP can produce greater bleed-through, thus needs to be exposed at a sub-optimal level. This is an internal reproduction of the Rok inhibition experiment.

b) The control values of Vinc/E-cad ratios vary considerably between Fig. 4D' and Fig. 4B', 4F'.

c) Similarly, the control levels of E-cad in Fig. 5A'-B' differ significantly. How do the authors explain this variation?

Response: The variation across different treatments can be easily explained by the experimental conditions. Depending on the experiments, the embryos may have different maternal and zygotic genotypes (detailed in, Methods: Fly lines and genetics), thus have a different proportion of tagged versus untagged protein pool. This will change the absolute values but not the relative changes. The entire protein pool is tagged in the case of the Rok inhibition experiment, thus the intensity values are highest. Entire maternal, but only half zygotic protein pool is tagged in Gα12/13 overexpression experiment (an F1 genetic cross), thus the intensity values are lower than the Rok inhibition experiment. Half of maternal as well as zygotic protein pool is tagged in RhoGEF2 RNAi experiment (an F2 genetic cross), thus the intensity values are about half of Rok inhibition experiment. Thus the proportion of fluorescently tagged protein pool is different; proportion being highest in Rok inhibition experiments and lowest in RhoGEF2-RNAi.

Further, this leads us to adjust/optimize imaging conditions for each treatment for simultaneous acquisition of two proteins; avoiding bleed through from GFP to mCherry channel, avoiding photo-bleaching in either channels and trying to get best signal possible. mCherry seems to bleach faster than GFP, thus imaging conditions need to be altered more for mCherry tagged protein as compared to GFP tagged ones. This makes it particularly difficult to compare Myosin-II intensities across treatments, as the presented intensity quantifications are mostly from mCherry tagged construct. Similar problem also exists for Vinculin-mCherry, and as a consequence also for Vinc/E-cad ratio. Since E-cadherin is GFP tagged, the imaging conditions could be kept most similar across treatments. This allows one to compare the E-cadherin

junctional intensities for controls in Figure 1F', 5A' and 5B' and to gauge the variability due to changes in the proportion of untagged protein pool.

Naturally, each treatment is accompanied with respective controls, which have the same proportion of untagged protein pool and are imaged under identical conditions. Thus, we can confidently attribute the changes between treatments and respective controls to the perturbation itself. Also, the quantifications are extremely robust when we present normalized intensities (Amplitude of polarity) as compared to mean intensities. E.g. compare the 'Amplitude of polarity' quantifications for junctional Myosin-II (controls in Figure 4A'', 4C'' and 4E'') as well as Vinc/E-cad ratio (controls in Figure 4B', 4D' and 4F'). Such normalization readily accounts for intensity changes due to changes in proportion of untagged proteins, as both numerator and denominator change proportionally. We have now tried to make this more explicit in the figure legends and have made appropriate connections with relevant 'methods' sub-sections.

Minor comments:

P. 6: the comment in square brackets was probably not meant to get to reviewers.

Response: Removed.

- Missing reference, page 10, last paragraph: "this condition is observed when adhesion is mildly compromised using partial knock-down of E-cadherin or α -catenin in the mesoderm and endoderm where medial Myosin-II accumulates at very high level"

Response: This line is now removed.

- Page 7, 8, 9: It is not clear what is meant with "uniform" loading

Response: In retrospect, using the term "uniform" is not informative enough and might even be misleading. We have now removed it and rephrased the sentences.

REVIEWERS' COMMENTS:

Reviewer #1 (Remarks to the Author):

Overall the authors have done a thorough job of addressing the reviewers' critiques. The additions to Supplemental Figure 1 and the more detailed description of the force balance calculations in the Methods Sections significantly improved the clarity of the measurements and the analyses. The additional justification of the correlation analyses improved the rigor. Their assumptions underlying models and analyses are now much clearer. The added experimental data also strengthened the manuscript. Overall, I think the manuscript is acceptable, with a few minor changes.

In reading this version, I identified a few issues that the authors should address

Lines 331-340: Their use of 'global' is misleading, and could be misinterpreted as referring to global expression levels. Instead, in lines 331-340 they are referring to (based on the figure reference) the cadherin levels at cell-cell junctions. However, in other cases, they explicitly overexpress cadherin and change the global expression levels (Lines 262-274; Lines 415 and following). This issue crops up throughout the manuscript (see line 372); they should be more precise.

Line 344: Their comment on shear "stretching the trans-cellular E-cadherin dimers" is not actually tested and probably incorrect on physical grounds. Both shear and normal forces would increase the force on cadherin bonds. I suggest (again) that the authors read Chang and Hammer (1996) "Influence of direction and type of applied force on the detachment of macromolecularly bound particles from surfaces", *Langmuir*, 12: 2271, as this paper directly demonstrates that shear or tangential forces are more effective than normal forces in disrupting bonds. The latter paper provides physical justification for some of their claims.

Line 334: 'led' not 'lead'

Reviewer #2 (Remarks to the Author):

The response of the authors to the reviewer comments is satisfactory overall, and the manuscript has been substantially improved. As far as I am concerned, only one issue remains:

In Fig.6A, cortical tensions are continuous at vertices (as claimed and as required to calculate shear stress) only in cells a and b, but not in c and d. This can be fixed when the same treatment as applied to T is also given to T1 and T3, as would be appropriate anyway. Thus, cortical tension should increase in cell d from the vertex outward along T1, and decrease in cell b on the other side of T1. And analogously at T3. Then the transfer of tensions is depicted to occur on all vertical junctions, as claimed and experimentally verified.

Reviewer #3 (Remarks to the Author):

In general, I am fine with most of the authors replies to the reviewers comments. My two central concerns are weakened but not fully eliminated:

- That the system is in equilibrium remains an assumption and one can easily imagine this not to be the case. The statement "Another justification for mechanical equilibrium is that the velocity of the cell and of vertices during development is much smaller than the recoil velocity after ablation, indicating that the unbalanced tensions at vertices are much smaller than junctional tensions" is a trivial one that does not provide a convincing foundation for the assumption.

- I remain somewhat critical about whether the central usage of inferred tension allows gathering sufficiently convincing evidence to make a very strong claim on force contributions. After all, in their response the authors first explain why recoil velocities are not a good measure and in the following they do state: "This indicates that inferred tension is at least as good a force estimate as recoil velocity". One could rephrase into as bad as.... Surely, both force estimates come with considerable pitfalls indeed.

Minor point:

Concerning usage of Spearman's versus Pearson's correlation: I would recommend adding the authors explanation to the methods section.

Altogether, I do appreciate that this approach provides new, solid and interesting data and comes up with a more distinct hypothesis on the contribution of the two actomyosin pools than previous work on this question. I trust that the field will take the conclusions with the necessary critical view should you decide to publish for which I have no major objection.

REVIEWERS' COMMENTS:

Reviewer #1 (Remarks to the Author):

Overall the authors have done a thorough job of addressing the reviewers' critiques. The additions to Supplemental Figure 1 and the more detailed description of the force balance calculations in the Methods Sections significantly improved the clarity of the measurements and the analyses. The additional justification of the correlation analyses improved the rigor. Their assumptions underlying models and analyses are now much clearer. The added experimental data also strengthened the manuscript. Overall, I think the manuscript is acceptable, with a few minor changes.

In reading this version, I identified a few issues that the authors should address

Lines 331-340: Their use of 'global' is misleading, and could be misinterpreted as referring to global expression levels. Instead, in lines 331-340 they are referring to (based on the figure reference) the cadherin levels at cell-cell junctions. However, in other cases, they explicitly overexpress cadherin and change the global expression levels (Lines 262-274; Lines 415 and following). This issue crops up throughout the manuscript (see line 372); they should be more precise.

Response: Thank you for pointing out this potential confusion. We have rephrased the relevant text, replacing 'global' with phrases like, overall, all junctions, average.

Line 344: Their comment on shear "stretching the trans-cellular E-cadherin dimers" is not actually tested and probably incorrect on physical grounds. Both shear and normal forces would increase the force on cadherin bonds. I suggest (again) that the authors read Chang and Hammer (1996) "Influence of direction and type of applied force on the detachment of macromolecularly bound particles from surfaces", *Langmuir*, 12: 2271, as this paper directly demonstrates that shear or tangential forces are more effective than normal forces in disrupting bonds. The latter paper provides physical justification for some of their claims.

Response: Thank you for suggesting this reference. We have included it in the 'results' as well as in the 'discussion' sections.

Line 334: 'led' not 'lead'

Response: Corrected. Thanks for spotting this.

Reviewer #2 (Remarks to the Author):

The response of the authors to the reviewer comments is satisfactory overall, and the manuscript has been substantially improved. As far as I am concerned, only one issue remains:

In Fig.6A, cortical tensions are continuous at vertices (as claimed and as required to calculate shear stress) only in cells a and b, but not in c and d. This can be fixed when the same treatment as applied to T is also given to T1 and T3, as would be appropriate anyway. Thus, cortical tension should increase in cell d from the vertex outward along T1, and decrease in cell b on the other side of T1. And analogously at T3. Then the transfer of tensions is depicted to occur on all vertical junctions, as claimed and experimentally verified.

Response: Thanks for spotting this. The error got introduced while preparing the figure panel. The real schematic was indeed how you describe it to be. The schematic is now corrected.

Reviewer #3 (Remarks to the Author):

In general, I am fine with most of the authors replies to the reviewers comments. My two central concerns are weakened but not fully eliminated:

- That the system is in equilibrium remains an assumption and one can easily imagine this not to be the case. The statement "Another justification for mechanical equilibrium is that the velocity of the cell and of vertices during development is much smaller than the recoil velocity after ablation, indicating that the unbalanced tensions at vertices are much smaller than junctional tensions" is a trivial one that does not provide a convincing foundation for the assumption.

Response: We agree with this point and indeed, we don't want to suggest that the tissue is in absolute mechanical equilibrium, in which case morphogenetic movements won't be possible. We argue that the system behaves close to mechanical equilibrium. We have clarified this in the manuscript Methods section. There are two chief elements in support of this hypothesis. The first is quoted by the reviewer and regards the respective speed of morphogenetic movement with respect to the speed of recoil velocity after ablation: the magnitude of forces driving morphogenetic movement is small compared to junctional tensions that are predominantly balanced at the vertices. The second argument is associated with the time scale separation between exchange kinetics of motors and actin filament dynamics (1-10s) and morphogenetic movement such as junction remodelling (in the order of 100s or more): molecular dynamics tend to equilibrate on the time scale of cell shape changes. We have now included additional explanation on these lines in 'methods: mechanical inference'.

- I remain somewhat critical about whether the central usage of inferred tension allows gathering sufficiently convincing evidence to make a very strong claim on force contributions. After all, in their response the authors first explain why recoil velocities are not a good measure and in the following they do state: "This indicates that inferred tension is at least as good a force estimate as recoil velocity". One could rephrase into as bad as.... Surely, both force estimates come with considerable pitfalls indeed.

Response: It is true that we are using 'inferred tension' as the primary estimate of

tension for performing correlational analysis. Though, we would like to emphasize that this is certainly not the only tension estimate. In fact, we are indirectly comparing Vinc/E-cad ratio and junctional Myosin-II, through inferred tension and recoil velocities. Our conclusions rely on use of mechanical inference and laser ablation experiments, as we can't estimate Vinc/E-cad ratio while also imaging Myosin-II. Such experiments, when possible, will remove the necessity of using mechanical inference and laser ablation as intermediary techniques to relate the distribution of Myosin-II (force generation) and Vinc/E-cad ratio (force sensing).

Minor point:

Concerning usage of Spearman's versus Pearson's correlation: I would recommend adding the authors explanation to the methods section.

Response: This is now included in the methods section.

Altogether, I do appreciate that this approach provides new, solid and interesting data and comes up with a more distinct hypothesis on the contribution of the two actomyosin pools than previous work on this question. I trust that the field will take the conclusions with the necessary critical view should you decide to publish for which I have no major objection.